# A solvable model of learning generative diffusion: theory and insights

**Hugo Cui**
Center of Mathematical Sciences and Applications,
Harvard University

**Cengiz Pehlevan**
Center for Brain Science,
Kempner Institute for the Study of Natural and Artificial Intelligence,
John A. Paulson School of Engineering and Applied Sciences,
Harvard University

**Yue M Lu**
John A. Paulson School of Engineering and Applied Sciences,
Center of Mathematical Sciences and Applications,
Harvard University

## Abstract

In this manuscript, we analyze a solvable model of flow or diffusion-based generative model. We consider the problem of learning a model parametrized by a two-layer auto-encoder, trained with online stochastic gradient descent, on a high-dimensional target density with an underlying low-dimensional manifold structure. We derive a tight asymptotic characterization of low-dimensional projections of the distribution of samples generated by the learned model, ascertaining in particular its dependence on the number of training samples. Building on this analysis, we discuss how mode collapse can arise, and lead to model collapse when the generative model is re-trained on generated synthetic data.

## Introduction

Diffusion and flow-based generative models represent a new paradigm in the sampling of high-dimensional probability densities. Such methods operate by recasting the sampling problem as a transport from a simple base distribution to the target density. The velocity field directing the transport can further be parameterized by a neural network, and learned from data [68, 70, 36, 34]. These ideas have been successfully implemented in a number of algorithmic frameworks [70, 48, 4, 49, 71], with applications ranging from image generation [55, 58, 63] to drug discovery [83].

The surprising effectiveness of such models in learning probability densities in high dimensions hints at the presence of *architectural biases* built in the network parametrization, placing strong priors on the class of densities generated by the model. When aligned with the target density, these architectural biases can allow generative models to learn a good approximation of the target from only a small number of training samples [35]. Naturally, when the biases are ill-suited to the task, they can also lead to poor solutions. Gaining a solid theoretical understanding on how the neural network architecture shapes the generated density is hence a central, yet still largely open, research question.

Arguably, the prominent technical obstruction lies in the need to reach a *precise* characterization of the density a given architecture learns to generate. A large fraction of theoretical studies of

39th Conference on Neural Information Processing Systems (NeurIPS 2025).

generative models [17, 14, 8, 40, 41, 43] analyze only the generative transport process, starting from the assumption that a $L^2$-accurate approximation of the velocity or score is available. This gap has in part been filled by a recent line of works [52, 15, 7, 13, 82], which establishes that some target densities can provably be learned by neural networks, provided sufficient width and number of samples. Such sample complexity bounds are however little descriptive of the shape of the generated density, nor do they capture *failure modes* where the architecture is not expressive enough to yield a good approximation. Closer to our work, authors of [21] conduct a tight analysis of a simple generative model, which is however limited to the highly stylized case of a binary Gaussian mixture target density with isotropic covariances. The present manuscript overcomes these barriers, and provides a *sharp* characterization of the generated distribution for models learning from a large class of non-trivially structured target densities.

**Main contributions**

We consider generative models parametrized by a two-layer Denoising Auto-Encoder (DAE) with tied weights and trainable skip connection, trained with online Stochastic Gradient Descent (SGD), in the framework of stochastic interpolation [4, 5]. We consider target distributions given by (possibly infinite) Gaussian mixtures in high dimensions, with generic cluster covariances, and centroids spanning a low-dimensional manifold – reflecting a pervasive intuition in machine learning [72, 84]. Overcoming previous limitations, we derive a *tight* asymptotic description of the generated distribution. More precisely,

- We provide a tight characterization of the training dynamics in terms of a set of deterministic Ordinary Differential Equations (ODEs), bearing over a finite set of low-dimensional summary statistics.

- Building on these results, we similarly provide a tight low-dimensional characterization of the generative transport process, thereby reaching a sharp description of low-dimensional projections of the generated density, as a function of the number of samples and sampling time. The theoretical predictions further quantitatively capture experiments on simple real datasets.

- We discuss and illustrate how a phenomenon akin to mode collapse [32] can arise, and lead to a loss of diversity in the generated density. Iterating and extending over our analysis to cases where the generated data is further re-used to train the generative model, we highlight how this mode collapse phenomenon can ultimately conduce to model collapse [66].

The code employed in this manuscript is accessible on this online repository.

**Related works**

**Sampling accuracy and dynamics–** A large body of works on diffusion-based and flow-based generative models has been devoted to the study of the generative process, assuming access to a $L^2$-accurate score estimate, or to the exact empirical score. [17, 14, 8, 40, 41, 43, 9, 16, 23, 42] provide rigorous bounds on appropriate distances between the target and generated probability distributions. The sequential emergence of structure in the transported density with sampling time has been investigated in [11, 10, 65, 64, 1, 79, 44, 28, 45], evidencing the presence of rich critical phenomena. [30] explore the computational hardness of sampling for an array of graphical models. This line of works, however, does not allow one to elucidate how such score estimates can be *learned from data* by practical architectures.

**Sample complexity bounds–** Complementing this set of results, a recent stream of work refined these bounds by further ascertaining the *sample complexity* of learning accurate score estimates [12, 38, 85, 25, 90, 33, 46, 29]. For data distributions close to the one considered in the present work, [27, 18] show how the score of Gaussian mixture densities can be learned algorithmically in efficient fashion. Similarly, for target densities with latent low-dimensional structure, [52, 15, 13] prove that DAE-parameterized models are able to learn the latent structure, and thus break the curse of dimensionality. [82] provide a full error analysis for models parametrized by deep ReLu networks. None of these bounds, however, allow for a precise elucidation of the *geometry* of the generated density. Furthermore, because such results primarily focus on settings where the target densities can be provably learned by the model with enough samples, they are not descriptive of unrealizable settings where the model is unable of perfect learning, and thus overlook possible failure modes and

biases. The present study on the other hand permits the exploration of the latter, and sheds light on mode and model collapse phenomena in the considered model.

**Tight characterization of learning in AEs–**  In order to study such cases, *sharper* results are therefore warranted. In this direction, a sizeable research effort has been devoted to analyzing the learning of AEs [77, 76], arguably the simplest instance of the class of denoiser neural networks used in generative models. The learning dynamics of AEs under (S)GD was characterized in [54] in the linear case, and [56] for non-linear models. [22] derive a tight asymptotic characterization of the learning of AEs for a denoising task, reaching a precise description of the generated density when this network is used to parametrize a generative model [21]. This characterization is however limited to a rather stylized target density, namely a binary Gaussian mixture with isotropic clusters, and thus fails to describe real data distributions. The present manuscript overcomes this barrier, and considers realistically structured target densities.

**Inductive bias in generative models–**  Even with moderately large training sets, generative models succeed in generating novel images, rather than reproducing memorized training samples [89, 88, 60, 53, 81, 47, 26, 80]. This surprising efficiency strongly hints at the presence of inductive biases inherited from the network parametrization, that nudge the model towards good solutions. The inductive bias of U-net architectures [59] has been investigated in [35], who observe how such architectures tend to learn adapted harmonic bases. In a similar spirit, the work of [37] evidences how the bias of such convolutional architectures towards learning equivariant and local scores helps to promote good solutions. Finally, [50] demonstrates that U-nets are closely related to message-passing algorithms on random hierarchical data, and thus particularly adapted to such structure. Complementing this line of works, the present manuscript offers insights on the bias of DAE architectures.

## 1  Problem formulation

We start by providing a succinct overview of the problem of sampling a target density $\rho$ over $\mathbb{R}^d$ using ideas from generative transport. For definiteness and ease of presentation, we consider in this manuscript the class of stochastic interpolant models [4], which shares substantial connections with other methods, including score-based diffusion models [71] and denoising methods [36, 34].

**Sampling–**  A sample $X_1 \sim \rho$ can be obtained from a Gaussian sample $X_0 \sim \mathcal{N}(0, \mathbb{I}_d)$ by evolving the latter for $t \in [0, 1]$ with the Stochastic Differential Equation (SDE)

$$\frac{dX_t}{dt} = \left( \dot{\beta}_t - \frac{\dot{\alpha}_t}{\alpha_t}\beta_t + \epsilon_t \frac{\beta_t}{\alpha_t^2} \right) f(t, X_t) + \left( \frac{\dot{\alpha}_t}{\alpha_t} - \frac{\epsilon_t}{\alpha_t^2} \right) X_t + \sqrt{2\epsilon_t}dW_t, \tag{1}$$

where $W$ is a Wiener process. This statement holds for any choice of interpolants $\alpha, \beta \in \mathcal{C}^2([0,1]), \epsilon \in \mathcal{C}^0([0,1])$ provided $\alpha(0) = \beta(1) = 1, \alpha(1) = \beta(0) = 0$ and $\alpha_t^2 + \beta_t^2 > 0, \epsilon_t \geq 0$ at all times $t \in [0, 1]$ [4]. In (1), the function $f : [0, 1] \times \mathbb{R}^d \to \mathbb{R}^d$ is defined as $f(t, x) = \mathbb{E}[x_1 | \alpha_t x_0 + \beta_t x_1 = x]$, with the conditional expectation bearing over $x_1 \sim \rho, x_0 \sim \mathcal{N}(0, \mathbb{I}_d)$. Intuitively, $f(t, x)$ can be interpreted as a *denoising function*, which aims to recover the sample $x_1$ from the interpolated version $\alpha_t x_0 + \beta_t x_1$, in which it is corrupted by the noise $x_0$.[1] Perhaps then unsurprisingly, the function $f$ admits a natural characterization as the minimizer of the quadratic denoising objective

$$\mathcal{R}[f] = \int\limits_0^1 \mathbb{E} \left\| f(t, \alpha_t x_0 + \beta_t x_1) - x_1 \right\|^2 dt. \tag{2}$$

This formulation provides an opportune pathway to learn the function $f$ governing the sampling (1) directly *from data*. The learning can be carried out following the usual machine learning rationale of (a) parametrizing $f$ by a denoiser neural network and (b) replacing the expectation in (2) by an empirical average over a training set.

---

[1]Note that the denoising function $f$ is related to the score function $s$ of the density of $\alpha_t x_0 + \beta_t x_1$ by the simple linear relation $\alpha_t^2 s(t, x) = \beta_t f(t, x) - x$.

**Architecture–** In the present manuscript, following [22], we consider the simplest instance of denoising neural network, namely a two-layer DAE

$$f_{b,w,v}(t,x) = b \times x + \frac{w}{\sqrt{d}}\sigma\left(\frac{w^\top x}{\sqrt{d}} + p_t v\right), \qquad (3)$$

with activation function $\sigma$, and trainable skip connection $b \in \mathbb{R}$. For simplicity, we assume that the decoder and encoder are *tied*, namely parametrized by a unique set of weights $w \in \mathbb{R}^{d \times r}$. This assumption allows for a more concise exposition of the technical results, and was not found to sensibly alter the phenomenology of the model. We discuss in Appendix A how the analysis can be extended to generically untied weights. In (3), $p : [0, 1] \to \mathbb{R}$ is an arbitrary time encoding scheme, and is embedded into the network preactivation through multiplication with a set of trainable weights $v \in \mathbb{R}^r$. Note that this is equivalent to including the time encoding $p_t$ as an extra dimension in the input $x$, and acting upon it with an extended set of $(d+1) \times r$ weights – with $v$ then corresponding to the first row thereof. Let us remark however that for the model (3), the inclusion of the time encoding scheme was not found to significantly alter the behaviour of the model in all probed settings.

Let us briefly situate the class of DAEs (3) with respect to models considered in related works. [15, 52] consider deep networks with ReLU activations, but require sparsity constraints on the weights. Closer to our work, [13] similarly assume shallow DAEs, but place themselves in the limit of infinite width ($r \to \infty$). On the opposite end of the spectrum, [21] consider shallow DAEs with a single hidden unit ($r = 1$), and sigmoidal activations $\sigma$. The architecture (3) considered in the present manuscript, on the other hand, corresponds to a more flexible class of finite-width networks with arbitrary activations. Let us note that while DAEs are amenable to theoretical characterization and consequently the main focus of the aforedescribed body of theoretical studies, they admittedly represent very simplified architectures when compared to e.g. U-Nets [59] used in practice. On the other hand, the model (3) retains the main architectural features of the latter, notably a bottleneck structure, a skip connection, and a time encoding scheme, and thus constitutes a valuable simplified model.

**Training–** The neural network (3) can now be used to parametrize the denoising function $f$ in the objective (2). We consider training the DAE (3) with *online (single-pass) SGD*, with learning rate $\eta$ and weight decay $\lambda$:

$$b_{\mu+1} - b_\mu = -\frac{\eta}{d^2}\left(\partial_b \mathbb{E}_t \left\|x_1^\mu - f_{b_\mu,w_\mu,v_\mu}(t, \alpha_t x_0^\mu + \beta_t x_1^\mu)\right\|^2\right), \qquad (4)$$

$$w_{\mu+1} - w_\mu = -\eta \nabla_w \mathbb{E}_t \left\|x_1^\mu - f_{b_\mu,w_\mu,v_\mu}(t, \alpha_t x_0^\mu + \beta_t x_1^\mu)\right\|^2 - \eta\frac{\lambda}{d}w_\mu. \qquad (5)$$

$$v_{\mu+1} - v_\mu = -\frac{\eta}{d}\nabla_v \mathbb{E}_t \left\|x_1^\mu - f_{b_\mu,w_\mu,v_\mu}(t, \alpha_t x_0^\mu + \beta_t x_1^\mu)\right\|^2 - \eta\frac{\lambda}{d}v_\mu. \qquad (6)$$

The expectation $\mathbb{E}_t$ over $t$ bears over the uniform distribution over $[0, 1]$, or any approximation thereof by a chosen set of points $\mathcal{G} = \{t_1, t_2, \dots\}$. The updates (4) are iterated $n$ times from a given initialization $b_0, w_0, v_0$. Note that in online SGD, a fresh pair of samples $x_1^\mu \sim \rho, x_0^\mu \sim \mathcal{N}(0, \mathbb{I}_d)$ is employed at each training step, and the number of training steps thus coincides with the number of samples. Finally, it will prove convenient to introduce the *training time* $\tau = {}^{2\eta n}/d$, a quantity that remains finite in the asymptotic limit considered, which we detail below. We accordingly denote by $b_\tau, w_\tau, v_\tau$ the values of the skip connection and weights at the end of training.

After training, the optimized DAE (3) $f_{b_\tau,w_\tau,v_\tau}$ can then be employed in the generative flow (1) as a proxy for the true denoising function $f$:

$$\frac{dX_t}{dt} = \left(\dot{\beta}_t - \frac{\dot{\alpha}_t}{\alpha_t}\beta_t + \epsilon_t\frac{\beta_t}{\alpha_t^2}\right)f_{b_\tau,w_\tau,v_\tau}(t, X_t) + \left(\frac{\dot{\alpha}_t}{\alpha_t} - \frac{\epsilon_t}{\alpha_t^2}\right)X_t + \sqrt{2\epsilon_t}dW_t. \qquad (7)$$

The SDE (7) can then be used for sampling. Because the learning is generically imperfect due to limited data and architectural bias, we generically have $f_{b_\tau,w_\tau,v_\tau} \neq f$, and thus the generated density $\hat{\rho}_\tau(t)$ –namely the law of $X_t$– differs from the target density $\rho$, even as $t \to 1$. One of the primary objectives of this work is to give a precise description of the discrepancy between $\rho$ and $\hat{\rho}_\tau(t)$ – *beyond* simply bounding probability distances therebetween–, through a sharp asymptotic characterization of $\hat{\rho}_\tau(t)$.

**Target density–** In this manuscript, we consider target densities given by a (possibly infinite) Gaussian mixture supported on a latent low-dimensional manifold. Namely,

$$\rho = \int_{\mathbb{R}^\kappa} \mathcal{N}(\mu(c), \Sigma(c)) d\pi(c). \tag{8}$$

In words, the centroids $\mu(c) \in \mathbb{R}^d$ of the different clusters lie on a $\kappa-$dimensional manifold, equipped with a coordinate system $c \in \mathbb{R}^\kappa$. The distribution of the clusters on the manifold is given by the relative weights $\pi(c)$. Finally, each cluster can exhibit a non-trivial covariance structure $\Sigma(c) \in \mathbb{R}^{d \times d}$. For the analysis, we further assume that all the covariances $\{\Sigma(c)\}_c$ are jointly diagonalizable, and admit well-defined spectral densities $\nu_c$ in the high-dimensional limit $d \to \infty$, leaving the generic case to future work. The density (8) reflects and models the celebrated *data manifold hypothesis* [72, 84], which posits that real data distributions lie on low-dimensional manifolds.

[52, 15, 13] similarly consider linear subspaces embedded in high dimensions, which can be viewed as special instances of (8) in the limit of vanishing covariances $\Sigma(c) = 0$ for all $c \in \mathbb{R}^\kappa$. Note also that the $K-$ modal Gaussian mixture distributions considered in e.g. [21, 86] correspond to the special case of (8) where the cluster distribution $\pi$ is a sum of $K$ Dirac deltas. Finally, the heavy-tailed distributions considered in [3, 2] correspond to the special case $\mu(c) = 0_d$ for all $c$.

**High-dimensional limit–** We aim at characterizing the generated density $\hat{\rho}_\tau(t)$ in the asymptotic limit of large data dimension and commensurably large number of samples, namely $d, n \to \infty$ with $n/d = \Theta_d(1)$. This asymptotic limit captures the non-trivial regime where the number of samples is not small enough for the neural network to trivially overfit, and conversely not infinite – thus allowing the investigation of finite data effects. We further suppose that the number of hidden units $r$ of the DAE and the intrinsic dimensionality of the data manifold $\kappa$, alongside all other parameters, remain finite: $r, \kappa, \lambda, \eta = \Theta_d(1)$. Moreover, the diameter of the mixture (8) is also supposed to remain finite, i.e. there exist $D = \Theta_d(1)$ such that $\|\mu(c)\|_2 \le D$ with probability 1 for $c \sim \pi$. Finally, we assume that the ambient dimension of the manifold is also finite, namely $\dim \mathrm{span}(\mu(c))_c = \Theta_d(1)$.

## 2 Precise characterization of the generated density

We are now in a position to detail our main technical findings, namely a sharp asymptotic characterization of the generated density $\hat{\rho}_\tau$ obtained from the sampling process (7), governed by the DAE $f_{b_\tau, w_\tau, v_\tau}$ (3) trained with online SGD (4). Because it is challenging to describe a high-dimensional probability distribution, we rather aim at characterizing *low-dimensional projections* thereof. Formally, let us fix a reference space $\mathcal{E} \subset \mathbb{R}^d$ with finite dimensionality $R = \Theta_d(1)$, and let $E \in \mathbb{R}^{d \times R}$ be a matrix whose columns form an orthonormal basis of $\mathcal{E}$. We aim at an sharp characterization of the law $\Pi_\mathcal{E} \hat{\rho}_\tau(t)$ of the projection $E^\top X_t$ of a generated sample $X_t \sim \hat{\rho}_\tau(t)$. In words, the subspace $\mathcal{E}$ corresponds to an observation space of interest, chosen by the statistician. Natural choices consists in electing a subspace where the target density exhibits non-trivial structure, with a view to probing how well it is reproduced at the level of the generated density – for example, the space spanned by the directions of larger variance of the target density $\rho$, or the space spanned by the latent manifold $\mathrm{span}(\{\mu(c)\}_c)$.

The derivation of this characterization proceeds in two steps. First, we derive a sharp asymptotic characterization of the high-dimensional training dynamics induced by SGD (4). More precisely, we describe the evolution of a set of *low-dimensional summary statistics* of the weights $w$ over training time in terms of a collection of limiting ODEs. These summary statistics encode all the geometric information on $w$ necessary to reach, in a second step, a sharp asymptotic characterization of the generative SDE (1) in terms of low-dimensional processes. Finally, this characterization yields a sharp description of $\Pi_\mathcal{E} \hat{\rho}_\tau(t)$ as a corollary.

### 2.1 Analysis of the learning

What weights are learned at the end of the training process (4), and how do they correlate with the main structural components of the target density, such as the centroids and directions of higher variance ? Answering this question is instrumental to subsequently build an understanding of how, during the generative process, the weight matrix shapes the generated density and allows to reproduce

the overall structure of the target density. This is the goal of our first result. On a technical level, it shows that the evolution under the SGD dynamics (4) of a set of summary statistics of the weights $w$ of the DAE (3) is asymptotically described by a collection of deterministic ODEs. These summary statistics importantly quantify the correlation of the trained DAE weights with the key geometric features of the target density.

**Result 2.1.** *(SGD dynamics) Let $\tau > 0$ denote the training time, and $w_\tau$ be the weight matrix obtained from the stochastic SGD dynamics* (4) *from an initialization $w_0$. The summary statistics $\mathcal{Q}_\tau \in \mathbb{R}^{r \times r}, Q_\tau : \mathbb{R}^\kappa \to \mathbb{R}^{r \times r}, G_\tau \in \mathbb{R}^{r \times R}, M_\tau : \mathbb{R}^\kappa \to \mathbb{R}^r$ defined as*

$$\mathcal{Q}_\tau = \frac{w_\tau^\top w_\tau}{d}, \qquad Q_\tau(c) = \frac{w_\tau^\top \Sigma(c) w_\tau}{d}, \qquad G_\tau = \frac{w_\tau^\top E}{\sqrt{d}}, \qquad M_\tau(c) = \frac{w_\tau^\top \mu(c)}{\sqrt{d}}, \quad (9)$$

*asymptotically concentrate in probability to the solutions at time $\vartheta = \tau$ of a system of coupled, finite-dimensional, deterministic ODEs*

$$\frac{d}{d\vartheta}\mathcal{Q}_\vartheta = F_\mathcal{Q}(O_\vartheta), \qquad \frac{d}{d\vartheta}Q_\vartheta = F_Q(O_\vartheta), \qquad \frac{d}{d\vartheta}G_\vartheta = F_G(O_\vartheta), \qquad \frac{d}{d\vartheta}M_\vartheta = F_M(O_\vartheta), \quad (10)$$

*using the shorthand $O_\vartheta = (\mathcal{Q}_\vartheta, Q_\vartheta, G_\vartheta, M_\vartheta, b_\vartheta)$ for more compact notation, although not all functions depend on the totality of the arguments in the collection $O_\vartheta$. Similarly, the time encoding weights $v_\tau$ converge to the solution of an ODE $d/d\vartheta v_\vartheta = F_v(O_\vartheta)$. The expression for the update functions $F_{\mathcal{Q},Q,G,M,v}$ is expounded in Appendix A. Finally, the value of the skip connection $b_\tau$ after training from an initialization $b_0$ is given by the compact closed-form expression*

$$b_\tau = \frac{\Lambda \mathbb{E}_t[\beta_t]}{\Lambda \mathbb{E}_t[\beta_t^2] + \mathbb{E}_t[\alpha_t^2]} \left[ 1 - e^{-\left(\Lambda \mathbb{E}_t[\beta_t^2] + \mathbb{E}_t[\alpha_t^2]\right)\tau} \right] + b_0 e^{-\left(\Lambda \mathbb{E}_t[\beta_t^2] + \mathbb{E}_t[\alpha_t^2]\right)\tau}, \quad (11)$$

*where we denoted $\Lambda$ the average covariance eigenvalue $\Lambda = \int d\pi(c) \int d\nu_c(\omega)\omega$.*

The derivation of Result 2.1, which we detail in Appendix A, follows the ideas of the seminal work of [61, 62], and is very close in spirit to the analysis of [56] for a related model, in the context of data reconstruction. It leverages the observation that in the high-dimensional limit, the SGD steps (4) self-average, and can be captured by a set of deterministic differential equations. These theoretical predictions accurately capture the dynamics observed in numerical experiments, as illustrated in Fig. 1. Finally, we detail for illustration in Appendix A the particular case of a linear DAE, for which the limiting ODEs (10) take simple, compact forms.

The ODEs (10) capture the evolution of the low-dimensional summary statistics $\mathcal{Q}, Q, G, M, P, T$, which subsume various geometric characteristics of the weight matrix $w_\tau$, over the SGD dynamics (4). Most notably, the statistic $M(c)$ measures the alignment between the weights $w$ with the centroids $\mu(c)$ that constitute the manifold (8), and thus measures how well the manifold structure is identified and learned. In the simple example of

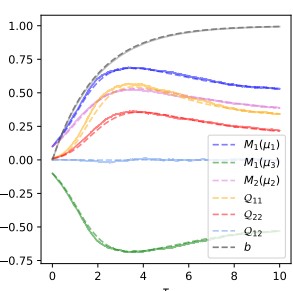

Figure 1: Evolution of the summary statistics $M_\tau, \mathcal{Q}_\tau$ and of the skip connection strength $b_\tau$ as a function of the training time $\tau$, for $\sigma = \tanh, r = 2, p_t = 0, \alpha_t = 1 - t, \beta_t = t, \mathcal{G} = \{1/2\}$. The target density is a trimodal Gaussian mixture $\rho = 1/3\mathcal{N}(\mu_1, I_d) + 1/3\mathcal{N}(\mu_2, I_d) + 1/3\mathcal{N}(\mu_3, I_d)$. Solid lines: numerical experiments in dimension $d = 1000$. Dashed: theoretical characterization (10) of Result 2.1.

a Gaussian mixture target density, $M(c)$ thus gives a quantitative sense of how well $w$ identifies and correlates with the different centroids, thereby learning the key structure of the density. The statistic $Q(c)$, on the other hand, gauges the alignment between the weights and the higher-variance directions of the density at point $c$ of the latent manifold. In the simple case of a unimodal target density with a spiked covariance, $Q(c)$ hence quantifies how well the spike is learned. Overall, the summary statistics appearing in Result 2.1 thus capture the correlation between the DAE weights $w$ and the key structural features of the target density (8), describing how well the latter are learned and encoded into the DAE parameters. The next section investigates how this encoded structure manifests during the generative transport, and allows in turn the reproduction of the structural elements in the generated density.

## 2.2 Analysis of the transport

We are now in a position to leverage the summary statistics $\mathcal{Q}_\tau, Q_\tau, \Theta_\tau, M_\tau, P, T$ characterized in Result 2.1 –which capture key geometric statistics of the weights $w_\tau$ of the DAE after training– to analyze the generative SDE (1). A crucial observation is that the SDE (1) is only non-linear in the $r-$dimensional subspace $\mathcal{W}_\tau = \mathrm{span}(w_\tau^i)_{i=1}^r$ spanned by the columns of the weight matrix $w_\tau$, and is on the other hand linear in the $d-r$ dimensional orthogonal subspace $\mathcal{W}_\tau^\perp$. The generative dynamics (1) of a sample $X_t$ can accordingly be compactly described by the linear evolution of $Y_t = \Pi_{\mathcal{W}_\tau}^\perp X_t$ (where we denoted $\Pi_{\mathcal{W}_\tau}^\perp$ the projection in $\mathcal{W}_\tau^\perp$), and the more complicated but finite-dimensional non-linear evolution of $Z_t = w_\tau^\top X_t/\sqrt{d} \in \mathcal{W}_\tau$. This statement is formalized in the following result:

**Result 2.2.** *(**generative dynamics**) Let $X_t$ be a stochastic process obeying the generative SDE (1) from an initialization $X_0 \sim \mathcal{N}(0, \mathbb{I}_d)$, and denote $Y_t = \Pi_{\mathcal{W}_\tau}^\perp X_t$ and $Z_t = w_\tau^\top X_t/\sqrt{d}$. Further define the shorthands $\Gamma_t = \dot{\beta}_t - \dot{\alpha}_t/\alpha_t \beta_t + \epsilon_t \beta_t/\alpha_t^2, \Delta_t^\tau = b_\tau \Gamma_t + \dot{\alpha}_t/\alpha_t - \epsilon_t/\alpha_t^2$, which depend on the schedule functions $\alpha, \beta, \epsilon$ and on the skip connection strength after training $b_\tau$. Then $Z_t$ obeys the low-dimensional SDE*

$$\frac{d}{dt} Z_t = \Delta_t^\tau Z_t + \Gamma_t \mathcal{Q}_\tau \sigma\left(Z_t + p_t v_\tau\right) + \sqrt{2\epsilon_t} \mathcal{Q}_\tau^{1/2} dB_t, \tag{12}$$

*from an initial condition $Z_0 \sim \mathcal{N}(0, \mathcal{Q}_\tau)$, with $B$ a $r-$dimensional Wiener process. On the other hand, $Y_t$ is independently Gaussian-distributed as*

$$Y_t \sim \mathcal{N}\left(0_{\mathcal{W}_\tau^\perp}, e^{2\int_0^t ds \Delta_s^\tau}\left[1 + 2\int_0^t e^{-2\int_0^s dh \Delta_h^\tau} \epsilon_s ds\right] \Pi_{\mathcal{W}_\tau}^\perp\right). \tag{13}$$

*The SDE (12) and equation (13) fully describe the law of $X_t$ in terms of low-dimensional quantities.*

We have thus reached a fully asymptotic characterization of the evolution of a sample $X_t$ transported by the SDE (1). Qualitatively, the density $\hat{\rho}_\tau(t)$ of $X_t$ is shaped in the column-space $\mathcal{W}_\tau$ of the weights $w_\tau$ by the action of the DAE network (second term in (12)), which acts as a drift term, while its scale is controlled by the contraction/expansion term $\Delta_t^\tau$ (first term in (12)), in which the skip connection $b_\tau$ intervenes. In $\mathcal{W}_\tau^\perp$, $\hat{\rho}_\tau(t)$ simply remains isotropic and Gaussian, with a time-varying variance succinctly given by (13). We remind that, from Result 2.1, the weights $w_\tau$ encode information on the key structural features of the target density $\rho$ (8). The corresponding drift term $\Gamma_t \mathcal{Q}_\tau \sigma\left(Z_t + p_t v_\tau\right)$ in (12) intuitively pushes the generated density along those important directions, ensuring that they can be approximately reproduced. This qualitative picture sheds light on the bias of the DAE-parametrized generative model (3). The DAE weights $w$ identify and learn important features of the target density from the dataset, allowing the model to implement at sampling time a non-trivial transport (12) in the corresponding space $\mathcal{W}_\tau$ to reproduce the main structural features of the target density. In the orthogonal subspace $\mathcal{W}_\tau^\perp$ on the other hand, the DAE is only able to rather crudely approximate the target by an isotropic Gaussian distribution (13), leveraging its skip connection $b$ to adjust the variance thereof.

## 2.3 Projected density

While Result 2.2 fully characterizes the distribution of $X_t$, the description is set the space $\mathcal{W}_\tau$, which changes with the training time $\tau$, rendering the result rather unwieldy. Ideally, we would like to transfer the characterization to a $\tau-$ independent reference space $E$, in order to allow for an easier comparison of the generated density at different training times — a comparison that will be subsequently discussed in Section 3, and illustrated in Figs. 2 and 3—. The desired technical result is thus a characterization of distribution of the projection $E^\top X_t$ of the generated sample $X_t$ in the reference space $\mathcal{E}$. We state this characterization in the following result.

**Corollary 2.3.** *(**Projected generated density**) The law of the projection $E^\top X_t$ of a sample $X_t$ in the space of interest $\mathcal{E}$ is given by*

$$E^\top X_t \overset{d}{=} G_\tau^\top \mathcal{Q}_\tau^+ Z_t + \mathcal{N}\left(0_R, e^{2\int_0^t ds \Delta_s^\tau}\left[1 + 2\int_0^t e^{-2\int_0^s dh \Delta_h^\tau} \epsilon_s ds\right]\left(\mathbb{I}_R - G_\tau^\top \mathcal{Q}_\tau^+ G_\tau\right)\right), \tag{14}$$

*where the law of $Z_t$ is characterized in Result 2.2 by the SDE (12), and the summary statistics $\mathcal{Q}_\tau, G_\tau$ are characterized in Result 2.1. $\mathcal{Q}_\tau^+$ denotes the Moore-Penrose pseudo-inverse of $\mathcal{Q}_\tau$.*

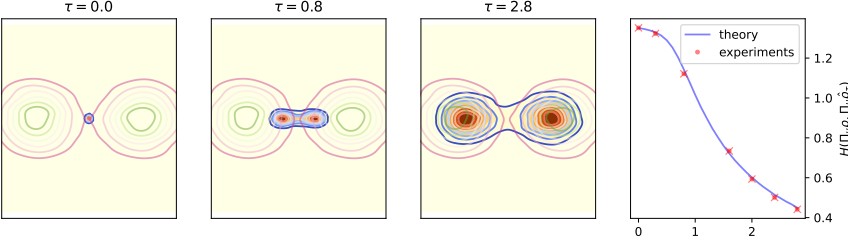

Figure 2: (**Left**) Evolution of the projected density $\Pi_{\mathcal{E}}\hat{\rho}_\tau$ generated by a DAE (3) with $r = 1$ hidden unit and $\sigma = \tanh$ activation, trained on a bimodal Gaussian mixture, with $\eta = 0.2, \lambda = 1.5, \epsilon_t = 0, p_t = 0, \alpha_t = 1 - t, \beta_t = t, \mathcal{G} = \{0.5\}$. The generative SDE (7) was run up to $t = 0.9$, and the subspace $\mathcal{E}$ is a plane containing the centroid of the target density. Different panels correspond to different training times $\tau$. Blue contours: contour levels of the theoretical prediction of Corollary 2.3 for the density $\Pi_{\mathcal{E}}\hat{\rho}_\tau$. Colormap: numerical experiments in large but finite dimension $d = 1000$. Green contours: contour levels of the target density $\rho$. (**Right**) Hellinger distance between the target and generated densities, projected in the space spanned by the centroid, as a function of the training time $\tau$.

Corollary 2.3 allows to transfer the characterization of Result 2.2, set in a training time dependent space $\mathcal{W}_\tau$, into a fixed, $\tau-$independent subspace $\mathcal{E}$. (14) reveals that the generated density is given by the sum of a non-trivial term $G_\tau^\top \mathcal{Q}_\tau^+ Z_t$, which captures the key learned structure of the target density, and a simple Gaussian term, which intuitively tries to approximate the remaining structural features which were not learned and reproduced. Let us finally remind that the choice of $\mathcal{E}$ is made by the statistician. To give more concrete examples, in the following, when considering Gaussian mixture targets, a natural choice for $\mathcal{E}$ is the space spanned by the cluster centroids. When dealing with unimodal targets with non-trivial covariance, we will choose the space spanned by the principal components.

## 3   Evolution of the generated density over training time

The theoretical characterizations of Results 2.1, 2.2 and 2.3 afford a complete characterization of low-dimensional projections of the generated density $\hat{\rho}_\tau$ as a function of the network architecture and training time $\tau$. They thereby provide a window to elucidate how the DAE architecture shapes the generated density. Importantly, the present analysis also allows the study of the evolution of the generated density over *training time $\tau$*, in addition to its evolution over sampling time $t$, which has traditionally been the focal point of past works, e.g. [11, 10, 17]. In the next paragraphs, we discuss these questions in the context of two examples, for a Gaussian mixture density and a real data distribution.

**Example 1 – Gaussian mixture**   We give as a first example the case of a Gaussian mixture target density $\rho$ with 2 isotropic modes. We consider a generative model parametrized by a DAE (3) with $r = 1$ hidden unit and $\sigma = \tanh$ activation. Fig. 2 illustrates, for different training times $\tau$, the generated density $\hat{\rho}_\tau$ projected in a two-dimensional space $\mathcal{E}$ containing the cluster centroids. A comparison between the theoretical predictions of Corollary 2.3 (blue contour levels) and numerical experiments in large but finite dimension $d = 1000$ (orange colormap) reveals a good agreement. Over training, the model successfully learns the data structure, and produces a generated density $\hat{\rho}_\tau$ exhibiting the correct bimodal structure in the relevant direction. Correspondingly, the estimated Hellinger distance between the generated and target densities in the subspace spanned by the cluster centroids monotonically decreases over training (Fig. 2 right). This simple example, which echoes the finding of e.g. [21], illustrates how the diffusion model (3) can successfully learn to reproduce simple target densities over training time. However, as mentioned above, the interest of deriving a precise characterization such as Corollary 2.3 primarily lies in the possibility to not only study settings where the model successfully learns the target density, but also settings where it fails and is biased – due to insufficient expressivity or data, or poor fit between the architecture and data structure. The following example instantiates such a case. We provide another example for a 3-modal Gaussian mixture, learned with a width $r = 4$, ReLU-activated DAE in Appendix B.

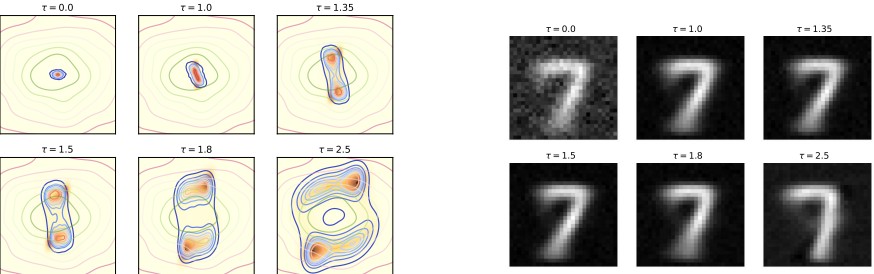

Figure 3: (**Left**) Evolution of the density $\Pi_{\mathcal{E}}\hat{\rho}_\tau$ generated by a DAE (3) with $r = 2$ hidden units and $\sigma = \tanh$ activation, trained on a Gaussian density with the MNIST sevens covariance, with $\eta = 0.2, \lambda = .784, \epsilon_t = p_t = 0, \alpha_t = 1 - t, \beta_t = t, \mathcal{G} = \{1/2\}$. The generative SDE (7) was run up to $t = 0.98$, and the subspace $\mathcal{E}$ is spanned by principal components of the target density. Different panels correspond to different training times $\tau$. Blue contours: contour levels of the theoretical prediction of Corollary 2.3 for the density $\hat{\rho}_\tau$. Colormap: numerical experiments. Green contours: contour levels of the target density $\rho$. (**Right**) Samples from the generated density $\hat{\rho}_\tau$, from a common initialization $X_0$ of the generative SDE (7), as a function of the training time $\tau$.

**Example 2– MNIST** In Fig. 3, a DAE-parametrized generative model with $r = 2$ hidden units and $\sigma = \tanh$ activation is trained to generate a Gaussian distribution with covariance matching that of MNIST sevens [39]. The generated probability $\hat{\rho}_\tau$ is represented in the principal two-dimensional eigenspace $\mathcal{E}$ of the MNIST sevens distribution. In a similar fashion to the first example, the generated density progressively adjusts to the shape of the target density (green and purple contours) over training time, first stretching in one direction into a bimodal density ($0 \lesssim \tau \lesssim 1.5$), with each mode being subsequently elongated ($\tau \gtrsim 1.8$), approaching the variance of the target in the secondary direction. This sequential emergence of directions of variance in the generated density has very visual consequences at the level of the generated images. Fig. 3 shows samples from the generated density $\hat{\rho}_\tau$ for varying training times $\tau$, transported from a common base sample $X_0$. For $0 \lesssim \tau \lesssim 1.5$, the generated image gains more resolution and becomes less noisy, as the first principal direction is learned. After $\tau \approx 1.8$, idiosyncratic features – such as the horizontal bar of the seven– emerge, as a second direction is learned, signaling increased diversity.

While the structure of the generated density $\hat{\rho}_\tau$ progressively adapts to that of the target $\rho$ over training, it retains a sizable discrepancy thereto – notably, the variance of $\hat{\rho}_\tau$ is markedly smaller. This dramatic reduction in variance betrays a detrimental bias in the model, which we more extensively explore in the next and last section. For completeness, we report in Fig. 10 in Appendix D an additional experiment on the FashionMNIST [87] dataset of clothing item images. The experiment again reveals a good agreement between the theoretical predictions and simulations (conducted this time directly on the original dataset, rather than a Gaussian approximation thereof), and a similar phenomenology.

## 4 Failure modes: mode(l) collapse

**Mode collapse–** The loss of variance in the modes of the generated density $\hat{\rho}_\tau$ when learning from the MNIST target distribution (Fig. 3) is reminiscent of the *mode collapse* phenomenon most commonly observed in the context of generative adversarial networks [32] and score-based models [24], and analyzed for variational inference in [69]. Mode collapse entails a loss in diversity in the generated images. In the present case, mode collapse is in fact a consequence of the unfitness of the architecture (3) for the target density at hand. To see this, first observe that the skip connection $b_\tau$ of the DAE model (3) contributes to increase the variance of the generated density $\hat{\rho}_\tau$, as it intervenes in the linear dilatation term $\Delta_t^\tau$ of the generative transport (12), see Result 2.2. In words, the stronger the strength $b_\tau$ of the skip connection, the more spread is the resulting generated density. However, note from (11) that the skip connection $b_\tau$ becomes of the same order as the average eigenvalue $\Lambda$ at large training times $\tau$. For real data distributions which tend to have a small number of large eigenvalues, and a large tail of small eigenvalues, $\Lambda$ is typically small, implying in turn a small $b_\tau$ and small variance for the modes of $\hat{\rho}_\tau$. Mode collapse causes a significant mismatch between the

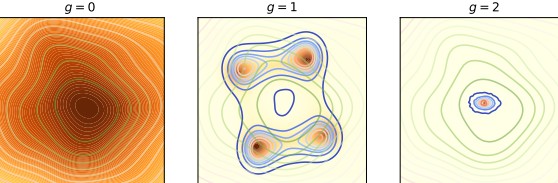

$g = 0 \qquad\qquad g = 1 \qquad\qquad g = 2$

Figure 4: (**left**) Target density $\rho$ corresponding to a Gaussian density equipped with the covariance of the distribution of MNIST sevens (**middle**) generated density $\Pi_{\mathcal{E}}\hat{\rho}_\tau^{(1)}$ (**right**) second generation density $\Pi_{\mathcal{E}}\hat{\rho}_\tau^{(2)}$ obtained by training the generative model (3) on the synthetic distribution $\hat{\rho}_\tau^{(1)}$. Blue contours: contour levels of the theoretical prediction of Corollary 2.3. Colormap: numerical experiments in large but finite dimension $d = 1000$. Green contours: contour levels of the target density $\rho$. At each successive generations, the same model specifications $\tau = 2.8, r = 2, \sigma = \tanh, \eta = 0.2, \lambda = .784, \epsilon_t = p_t = 0, \alpha_t = 1-t, \beta_t = t, \mathcal{G} = \{1/2\}$ were employed. The generative SDEs (7) were run up to $t = 0.98$ at reach generation. Finally, the subspace $\mathcal{E}$ is spanned by principal components of the target density.

generated distribution $\hat{\rho}_\tau$ and the target distribution $\rho$. A possible remedy would be to fix the skip connection strength $b$, instead of training it, ensuring it retains a sufficiently large value. We briefly present additional numerical experiments in Appendix D.

**Model collapse–** This mismatch can be further aggravated when the biased synthetic data thus generated is *re-used* to train another generative model, with successive generations of generated densities $\hat{\rho}_{(1),\tau_1}, \hat{\rho}_{(2),\tau_2}, ...$ becoming increasingly biased. This rapid degradation is an instantiation of the *model collapse* phenomenon –described in [66] and analyzed in [91] for simple linear single-step denoising models– can be opportunely analyzed for non-linear diffusion-based models within the present theoretical framework. To see this, observe that training the generation $g + 1$ model on data produced by the generation $g$ model corresponds to adopting the density $\hat{\rho}_{(g),\tau_g}$ generated by the latter as the target distribution. Seasonably, at each generation $\hat{\rho}_{(g),\tau_g}$ further remains of the form (8), and Results 2.1, 2.2 and 2.3 thus apply. This observation is formalized in the following remark.

**Remark 4.1.** (***Successive generated densities***) *At each generation, $\hat{\rho}_{(g),\tau_g}$ is again of the form* (8)*, with $\kappa = r, \mu(c) = c$ and*

$$\pi = \Pi_{\mathcal{W}_{(g),\tau_g}}\hat{\rho}_{(g),\tau_g}, \quad \forall c \in \mathbb{R}^\kappa, \ \Sigma(c) = e^{2\int_0^t ds \Delta_s^\tau}\left[1 + 2\int_0^t e^{-2\int_0^s dh \Delta_h^\tau}\epsilon_s ds\right]\Pi_{\mathcal{W}_{(g),\tau_g}}^\perp. \quad (15)$$

*In* (15)*, $\Pi_{\mathcal{W}_{(g)\tau_g}}\hat{\rho}_{(g),\tau_g}$, $\Delta^{\tau_g}$ are given by Results 2.1, 2.2 and 2.3 evaluated for a target density $\hat{\rho}_{(g-1),\tau_{g-1}}$. We remind that $\mathcal{W}_{(g),\tau_g}$ denotes the column-space of the trained weights of the generation $g$ model.*

In words, Remark 4.1 ensures that one can apply Results 2.1, 2.2 and 2.3 to iteratively reach a characterization of $\hat{\rho}_{(g+1),\tau_{g+1}}$ from that of $\hat{\rho}_{(g),\tau_g}$. Fig. 4 illustrates the first two generations of synthetic distributions $\hat{\rho}_{(1),\tau_1}, \hat{\rho}_{(2),\tau_2}$, when the initial target distribution $\rho$ is given by the same Gaussian approximation of MNIST sevens considered in Fig. 3. The mode collapse phenomenon described in the previous subsection, and apparent at generation $g = 1$, gets aggravated upon re-training. As a result, the density $\hat{\rho}_{(2),\tau_2}$ generated at $g = 2$ exhibits smaller even variance, and is devoid of meaningful structure.

## Conclusions

In this manuscript, we provide a precise and exhaustive theoretical characterization of the distribution of the samples generated by a simple diffusion model. Considering a model parametrized by a shallow DAE, in the limit of high dimensions, we derive a tight characterization of low-dimensional projections of the generated density, ascertaining in particular its evolution over training. Our results, which also describe tasks with realistically structured target densities, capture how the architectural bias can lead to mode collapse, and ultimately model collapse if the synthetic data is re-used for training.

**Acknowledgements**

The authors would like to thank Michael Albergo, Binxu Wang, Sabarish Sainathan, Lenka Zdeborová, and Emanuele Troiani for insightful discussions. HC acknowledges support from the Center of Mathematical Sciences and Applications (CMSA) of Harvard University. CP is supported by NSF Award DMS-2134157, NSF CAREER Award IIS-2239780, and a Sloan Research Fellowship. YML is supported by the Harvard FAS Dean's Fund for Promising Scholarship, and by a Harvard College Professorship. This work has been made possible in part by a gift from the Chan Zuckerberg Initiative Foundation to establish the Kempner Institute for the Study of Natural and Artificial Intelligence.

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

# A Derivation of Result 2.1

In this Appendix, we detail the derivation of the tight ODE description (10) of the SGD training dynamics (4), as provided in Result 2.1. We sequentially examine the dynamics for the skip connection $b$ and the weight matrix $w$.

## A.1 SGD dynamics of the skip connection

We first derive a closed-form expression for the evolution of the skip connection strength $b$ (3) over the SGD iterations. We recall that the latter read

$$b_{\mu+1} - b_\mu = -\frac{\eta}{d^2}\mathbb{E}_t\left[-2(1-b\beta_t)\beta_t\|x_1^\mu\|^2 + 2b\alpha_t^2\|x_0^\mu\|^2 + O(\sqrt{d})\right], \tag{16}$$

keeping only leading order terms. Note that $\|x_1^\mu\|^2/d$ (resp. $\|x_0^\mu\|^2/d$) asymptotically concentrate to $\Lambda$ (resp. 1) in the limit $d \to \infty$. Therefore, the increment $db = b_{\mu+1} - b_\mu$ self-averages as

$$\frac{d}{2\eta}db = \mathbb{E}_t\left[\beta_t(1-b\beta_t)\Lambda - b\alpha_t^2\right]. \tag{17}$$

The ODE (17) gives in closed-form the evolution of the skip connection strength over the SGD training (4), in the considered high-dimensional limit. Note that interestingly, while the training algorithm is inherently stochastic, the randomness averages out in the high-dimensional limit, leading the dynamics to concentrate to a deterministic limiting ODE.

## A.2 SGD dynamics of the weight matrix

### A.2.1 SGD update

We now turn to deriving a similar tight asymptotic characterization for the evolution of the weight matrix $w$ (3) under the SGD dynamics. Let us first write explicitly the SGD updates (4). Developing the derivative, and dropping the time index $\mu$ for readability, for $1 \le \gamma \le r, 1 \le i \le d$, the SGD update of the weight matrix reads

$$dw_{i\gamma} = -\frac{2\eta}{d}\sum_{\delta=1}^r \mathbb{E}_t\left[\sigma(\omega_\gamma^t + v_\gamma p_t)\sigma(\omega_\delta^t + v_\delta p_t)\right]w_{i\delta} + \frac{2\eta}{\sqrt{d}}\mathbb{E}_t\left[((1-b\beta_t)x_i^1 - b\alpha_t x_i^0)\sigma(\omega_\gamma^t + v_\gamma p_t)\right]$$

$$-\frac{2\eta}{\sqrt{d}}\mathbb{E}_t\left[(\alpha_t x_i^0 + \beta_t x_i^1)\sum_{\delta=1}^r \sigma(\omega_\delta^t + v_\delta p_t)\left(\frac{w_\delta^\top w_\gamma}{d}\right)\sigma'(\omega_\gamma^t + v_\gamma p_t)\right]$$

$$+\frac{2\eta}{\sqrt{d}}\mathbb{E}_t\left[(\alpha_t x_i^0 + \beta_t x_i^1)((1-b\beta_t)\lambda_\gamma^1 - b\alpha_t\lambda_\gamma^0)\sigma'(\omega_\gamma^t + v_\gamma p_t)\right] - \frac{\eta}{d}\lambda w_{i\gamma} \tag{18}$$

We introduced the shorthands

$$\lambda_\gamma^1 \equiv \frac{w_\gamma^\top x^1}{\sqrt{d}}, \qquad \lambda_\gamma^0 \equiv \frac{w_\gamma^\top x^0}{\sqrt{d}}, \qquad \omega_\gamma^t = \alpha_t\lambda_\gamma^0 + \beta_t\lambda_\gamma^1. \tag{19}$$

### A.2.2 Expected increment

In the likeness of the settings studied by e.g. [61, 62, 56], we expect the dynamics to asymptotically self-average. Let us accordingly evaluate the expectation $\mathbb{E}[dw_{i\gamma}]$ over the running data sample $x_\mu^{1,0}$. This can be compactly rewritten as

$$\mathbb{E}[dw_{i\gamma}] = \mathbb{E}_t\mathbb{E}_c[\mathbb{E}^c dw_{i\gamma}^{t,c}], \tag{20}$$

with the expectation $\mathbb{E}_c$ bearing over the manifold coordinate $c \sim \pi$ (8). Conditional on the manifold coordinate $c$, the expectation $\mathbb{E}^c$ bears over the Gaussian random variable associated to the $c-$indexed cluster in (8), distributed as $\mathcal{N}(\mu(c), \Sigma(c))$. We remind the reader that the covariances $\{\Sigma(c)\}_c$ are assumed jointly diagonalizable. Without loss of generality, we place ourselves in the basis in which

they are directly diagonal, and denote in the following by $\varrho_i^c$ the $i-$th eigenvalue of $\Sigma(c)$. Taking the expectation over (18), the expected increment then reads

$$
\begin{aligned}
\mathbb{E}^c[dw_{i\gamma}^{t,c}] = & -\frac{2\eta}{d}\sum_{\delta=1}^{r}\underbrace{\mathbb{E}^c\left[\sigma(\omega_\gamma^t + v_\gamma p_t)\sigma(\omega_\delta^t + v_\delta p_t)\right]}_{\mathcal{A}_{\gamma\delta}^{t,c}}w_{i\delta} + \frac{2\eta}{\sqrt{d}}(1-b\beta_t)\underbrace{\mathbb{E}^c\left[x_i^1\sigma(\omega_\gamma^t + v_\gamma p_t)\right]}_{\mathcal{B}_{i\gamma}^{1,t,c}} \\
& -\frac{2\eta}{\sqrt{d}}b\alpha_t\underbrace{\mathbb{E}^c\left[x_i^0\sigma(\omega_\gamma^t + v_\gamma p_t)\right]}_{\mathcal{B}_{i\gamma}^{0,t,c}} \\
& -\frac{2\eta}{\sqrt{d}}\alpha_t\sum_{\delta=1}^{r}\left(\frac{w_\delta^\top w_\gamma}{d}\right)\underbrace{\mathbb{E}^c\left[x_i^0\sigma(\omega_\delta^t + v_\delta p_t)\sigma'(\omega_\gamma^t + v_\gamma p_t)\right]}_{\mathcal{C}_{i\gamma\delta}^{0,t,c}} \\
& -\frac{2\eta}{\sqrt{d}}\beta_t\sum_{\delta=1}^{r}\left(\frac{w_\delta^\top w_\gamma}{d}\right)\underbrace{\mathbb{E}^c\left[x_i^1\sigma(\omega_\delta^t + v_\delta p_t)\sigma'(\omega_\gamma^t + v_\gamma p_t)\right]}_{\mathcal{C}_{i\gamma\delta}^{1,t,c}} \\
& +\frac{2\eta}{\sqrt{d}}\alpha_t(1-b\beta_t)\underbrace{\mathbb{E}^c[x_i^0\lambda_\gamma^1\sigma'(\omega_\gamma^t + v_\gamma p_t)]}_{\mathcal{D}_{i\gamma}^{01,t,c}} - \frac{2\eta}{\sqrt{d}}\alpha_t^2 b\underbrace{\mathbb{E}^c[x_i^0\lambda_\gamma^0\sigma'(\omega_\gamma^t + v_\gamma p_t)]}_{\mathcal{D}_{i\gamma}^{00,t,c}} \\
& +\frac{2\eta}{\sqrt{d}}\beta_t(1-b\beta_t)\underbrace{\mathbb{E}^c[x_i^1\lambda_\gamma^1\sigma'(\omega_\gamma^t + v_\gamma p_t)]}_{\mathcal{D}_{i\gamma}^{11,t,c}} - \frac{2\eta}{\sqrt{d}}\alpha_t\beta_t b\underbrace{\mathbb{E}^c[x_i^1\lambda_\gamma^0\sigma'(\omega_\gamma^t + v_\gamma p_t)]}_{\mathcal{D}_{i\gamma}^{10,t,c}} - \frac{2\eta}{d}\lambda w_{i\gamma}.
\end{aligned}
$$

$$(21)$$

The various coefficients $\mathcal{A}^{t,c}, \mathcal{B}^{t,c}, \mathcal{C}^{t,c}, \mathcal{D}^{t,c}$ can be computed leveraging the fact that the data components $x_i^{1,0}$ are weakly correlated with the local fields $\omega, \lambda^{0,1}$, i.e. have $\Theta_d(1/\sqrt{d})$ covariance. Using the expansions for weakly correlated Gaussian variables reported e.g. in [57] (Appendix B.1),

we reach

$$\mathcal{A}^{t,c}_{\gamma\delta} = I^{t,c}_{\sigma\sigma}(\gamma,\delta) \tag{22}$$

$$\mathcal{B}^{1,t,c}_{\gamma} = I^{t,c}_{\sigma}(\gamma)\mu^c_i + \frac{1}{\sqrt{d}}\frac{\beta_t w_{i\gamma}\varrho^c_i}{\Omega^{t,c}_{\gamma\gamma}}(I^{t,c}_{\sigma\omega}(\gamma,\gamma) - \beta_t M^c_\gamma I^{t,c}_{\sigma}(\gamma)) \tag{23}$$

$$\mathcal{B}^{0,t,c}_{\gamma} = \frac{1}{\sqrt{d}}\frac{\alpha_t w_{i\gamma}}{\Omega^{t,c}_{\gamma\gamma}}(I^{t,c}_{\sigma\omega}(\gamma,\gamma) - \beta_t M^c_\gamma I^{t,c}_{\sigma}(\gamma)) \tag{24}$$

$$\mathcal{C}^{0,t,c}_{i\gamma\delta} = \frac{1 - \delta_{\gamma\delta}}{\sqrt{d}}\frac{\alpha_t}{\Omega^{t,c}_{\gamma\gamma}\Omega^{t,c}_{\delta\delta} - (\Omega^{t,c}_{\gamma\delta})^2}\left[\left(I^{t,c}_{\sigma'\sigma\omega}(\gamma,\delta,\gamma) - \beta_t M^c_\gamma I^{t,c}_{\sigma'\sigma}(\gamma,\delta)\right)(\Omega^{t,c}_{\delta\delta}w_{i\gamma} - \Omega^{t,c}_{\gamma\delta}w_{i\delta})\right.$$
$$\left. + \left(I^{t,c}_{\sigma'\sigma\omega}(\gamma,\delta,\delta) - \beta_t M^c_\delta I^{t,c}_{\sigma'\sigma}(\gamma,\delta)\right)(\Omega^{t,c}_{\gamma\gamma}w_{i\delta} - \Omega^{t,c}_{\gamma\delta}w_{i\gamma})\right]$$
$$+ \frac{\delta_{\gamma\delta}}{\Omega^{t,c}_{\gamma\gamma}}\alpha_t w_{i\gamma}(I^{t,c}_{\sigma'\sigma\omega}(\gamma,\gamma,\gamma) - \beta_t M^c_\gamma I^{t,c}_{\sigma'\sigma}(\gamma,\gamma)) \tag{25}$$

$$\mathcal{C}^{1,t,c}_{i\gamma\delta} = I^{t,c}_{\sigma'\sigma}(\gamma,\delta)\mu^c_i + \frac{\beta_t \varrho^c_i}{\alpha_t}\mathcal{C}^{0,t,c}_{i\gamma\delta} \tag{26}$$

$$\mathcal{D}^{01,t,c}_{i\gamma} = \frac{1}{\sqrt{d}}\frac{Q^c_{\gamma\gamma}\alpha_t w_{i\gamma}}{Q^c_{\gamma\gamma}\Omega^{t,c}_{\gamma\gamma} - \beta^2_t(Q^c_{\gamma\gamma})^2}\left[I^{t,c}_{\lambda^1\sigma'\omega}(\gamma,\gamma,\gamma) - \beta_t I^{t,c}_{(\lambda^1)^2\sigma'}(\gamma,\gamma)\right] \tag{27}$$

$$\mathcal{D}^{00,t,c}_{i\gamma} = \frac{1}{\sqrt{d}}\frac{w_{i\gamma}}{\mathcal{Q}_{\gamma\gamma}\Omega^{t,c}_{\gamma\gamma} - \alpha^2_t(\mathcal{Q}_{\gamma\gamma})^2}\left[I^{t,c}_{(\lambda^0)^2\sigma'}(\gamma,\gamma)(\Omega^{t,c}_{\gamma\gamma} - \alpha^2_t\mathcal{Q}_{\gamma\gamma})\right] \tag{28}$$

$$\mathcal{D}^{10,t,c}_{i\gamma} = \mu^c_i I^{t,c}_{\lambda^0\sigma'}(\gamma,\gamma) + \frac{1}{\sqrt{d}}\frac{\mathcal{Q}_{\gamma\gamma}\beta_t\varrho^c_i w_{i\gamma}}{\mathcal{Q}_{\gamma\gamma}\Omega^{t,c}_{\gamma\gamma} - \alpha^2_t(\mathcal{Q}_{\gamma\gamma})^2}\left[I^{t,c}_{\lambda^0\sigma'\omega}(\gamma,\gamma,\gamma) - \alpha_t I^{t,c}_{(\lambda^0)^2\sigma'}(\gamma,\gamma) - M^c_\gamma\beta_t I^{t,c}_{\lambda^0\sigma'}(\gamma,\gamma)\right] \tag{29}$$

$$\mathcal{D}^{11,t,c}_{i\gamma} = \mu^c_i I^{t,c}_{\lambda^1\sigma'}(\gamma,\gamma) + \frac{1}{\sqrt{d}}\frac{w_{i\gamma}\varrho^c_i}{Q^c_{\gamma\gamma}\Omega^{t,c}_{\gamma\gamma} - \beta^2_t(Q^c_{\gamma\gamma})^2}\left[(I^{t,c}_{(\lambda^1)^2\sigma'}(\gamma,\gamma) - M^c_\gamma I^{t,c}_{\lambda^1\sigma'}(\gamma,\gamma))(\Omega^{t,c}_{\gamma\gamma} - \beta^2_t Q^c_{\gamma\gamma})\right]. \tag{30}$$

We introduced the summary statistics

$$M^c = \frac{w^\top\mu(c)}{\sqrt{d}}, \qquad Q^c = \frac{w^\top\Sigma(c)w}{d}, \tag{31}$$

$$\mathcal{Q} = \frac{w^\top w}{d}, \qquad \Omega^{t,c} = \alpha^2_t\mathcal{Q} + \beta^2_t Q^c, \qquad T^{ck} = \mu(c)^\top\mu(k) \tag{32}$$

One also needs to introduce the further statistics

$$G = \frac{w^\top E}{\sqrt{d}}, \qquad P^c = E^\top\mu(c), \tag{33}$$

where we remind that the columns of $E \in \mathbb{R}^{d\times R}$ constitue an orthonormal basis of the $R-$dimensional subspace $\mathcal{E}$ in which we aim to characterize the generated density. Finally, we also used the shorthands:

$$I^{t,c}_{\sigma\sigma}(\gamma,\delta) = \mathbb{E}_{\omega_\gamma,\omega_\delta}[\sigma(\omega_\gamma + v_\gamma p_t)\sigma(\omega_\delta + v_\delta p_t)], \qquad \omega_\gamma,\omega_\delta \sim \mathcal{N}\left(\beta_t M^c_{(\gamma,\delta)},\Omega^{t,c}_{(\gamma,\delta)}\right) \tag{34}$$

$$I^{t,c}_{\sigma}(\gamma) = \mathbb{E}_{\omega_\gamma}[\sigma(\omega_\gamma + v_\gamma p_t)], \qquad \omega_\gamma \sim \mathcal{N}\left(\beta_t M^c_\gamma,\Omega^{t,c}_{\gamma\gamma}\right) \tag{35}$$

$$I^{t,c}_{\sigma\omega}(\gamma,\delta) = \mathbb{E}_{\omega_\gamma,\omega_\delta}[\sigma(\omega_\gamma + v_\gamma p_t)\omega_\delta], \qquad \omega_\gamma,\omega_\delta \sim \mathcal{N}\left(\beta_t M^c_{(\gamma,\delta)},\Omega^{t,c}_{(\gamma,\delta)}\right) \tag{36}$$

$$I^{t,c}_{\sigma'\sigma\omega}(\gamma,\delta,\epsilon) = \mathbb{E}_{\omega_\gamma,\omega_\delta,\omega_\epsilon}[\sigma'(\omega_\gamma + v_\gamma p_t)\sigma(\omega_\delta + v_\delta p_t)\omega_\epsilon], \qquad \omega_\gamma,\omega_\delta,\omega_\epsilon \sim \mathcal{N}\left(\beta_t M^c_{(\gamma,\delta,\epsilon)},\Omega^{t,c}_{(\gamma,\delta,\epsilon)}\right) \tag{37}$$

$$I^{t,c}_{\sigma'\sigma}(\gamma,\delta) = \mathbb{E}_{\omega_\gamma,\omega_\delta}[\sigma'(\omega_\gamma + v_\gamma p_t)\sigma(\omega_\delta + v_\delta p_t)], \qquad \omega_\gamma,\omega_\delta \sim \mathcal{N}\left(\beta_t M^c_{(\gamma,\delta)},\Omega^{t,c}_{(\gamma,\delta)}\right), \tag{38}$$

and

$$I^{t,c}_{\lambda^1 \sigma' \omega}(\gamma, \delta, \epsilon) = \mathbb{E}_{\lambda^1_\gamma, \omega_\delta, \omega_\epsilon}[\lambda^1_\gamma \sigma'(\omega_\delta + v_\delta p_t)\omega_\epsilon],$$

$$\lambda^1_\gamma, \omega_\delta, \omega_\epsilon \sim \mathcal{N}\left(\begin{pmatrix} M^c_\gamma \\ \beta_t M^c_{(\delta, \epsilon)} \end{pmatrix}, \begin{pmatrix} Q^c_{\gamma\gamma} & \beta_t Q^c_{\gamma, (\delta, \epsilon)} \\ \beta_t (Q^c_{\gamma, (\delta, \epsilon)})^\top & \Omega^{t,c}_{(\delta, \epsilon)} \end{pmatrix}\right) \tag{39}$$

$$I^{t,c}_{(\lambda^1)^2 \sigma'}(\gamma, \delta) = \mathbb{E}_{\omega_\gamma, \omega_\delta}[(\lambda^1_\gamma)^2 \sigma'(\omega_\delta + v_\delta p_t)],$$

$$\lambda^1_\gamma, \omega_\delta \sim \mathcal{N}\left(\begin{pmatrix} M^c_\gamma \\ \beta_t M^c_\delta \end{pmatrix}, \begin{pmatrix} Q^c_{\gamma\gamma} & \beta_t Q^c_{\gamma, \delta} \\ \beta_t (Q^c_{\gamma, \delta}) & \Omega^{t,c}_\delta \end{pmatrix}\right) \tag{40}$$

$$I^{t,c}_{\lambda^0 \sigma' \omega}(\gamma, \delta, \epsilon) = \mathbb{E}_{\lambda^1_\gamma, \omega_\delta, \omega_\epsilon}[\lambda^0_\gamma \sigma'(\omega_\delta + v_\delta p_t)\omega_\epsilon],$$

$$\lambda^0_\gamma, \omega_\delta, \omega_\epsilon \sim \mathcal{N}\left(\begin{pmatrix} 0 \\ \beta_t M^c_{(\delta, \epsilon)} \end{pmatrix}, \begin{pmatrix} \mathcal{Q}_{\gamma\gamma} & \alpha_t \mathcal{Q}_{\gamma, (\delta, \epsilon)} \\ \alpha_t (\mathcal{Q}_{\gamma, (\delta, \epsilon)})^\top & \Omega^{t,c}_{(\delta, \epsilon)} \end{pmatrix}\right) \tag{41}$$

$$I^{t,c}_{(\lambda^0)^2 \sigma'}(\gamma, \delta) = \mathbb{E}_{\omega_\gamma, \omega_\delta}[(\lambda^0_\gamma)^2 \sigma'(\omega_\delta + v_\delta p_t)],$$

$$\lambda^0_\gamma, \omega_\delta \sim \mathcal{N}\left(\begin{pmatrix} 0 \\ \beta_t M^c_\delta \end{pmatrix}, \begin{pmatrix} \mathcal{Q}_{\gamma\gamma} & \alpha_t \mathcal{Q}_{\gamma, \delta} \\ \alpha_t (\mathcal{Q}_{\gamma, \delta}) & \Omega^{t,c}_\delta \end{pmatrix}\right) \tag{42}$$

$$\tag{43}$$

Let us briefly summarize at this point the derivation so far. Starting from the SGD update on $w$, we showed that the expected increment can be written solely in terms of a small number summary statistics $M, Q, \mathcal{Q}, \Omega, T$, in addition to the weight components themselves. Next, we aim at closing these equations on the summary statistics, thus reaching a concise low-dimensional description of the SGD dynamics.

### A.2.3 Update equation for the summary statistics

The training dynamics of the DAE weights $w$ are thus governed by set of finite-dimensional summary statistics. To close the equations and reach a self-contained characterization, we now turn to deriving the induced dynamics of the summary statistics. To that end, following e.g. [31], it proves convenient to first introduce a new set of summary statistics *densities*. For any $\varrho : \mathbb{R}^\kappa \to \mathbb{R}$ –denoting a joint sequence of eigenvalues $\{\varrho(c)\}$–, let us assume the existence of the densities $m, p : \mathbb{R}^\kappa \times \mathcal{F}(\mathbb{R}^\kappa, \mathbb{R}) \to \mathbb{R}, q, g : \times \mathcal{F}(\mathbb{R}^\kappa, \mathbb{R}) \to \mathbb{R}$ and $\theta : (\mathbb{R}^\kappa)^2 \times \mathcal{F}(\mathbb{R}^\kappa, \mathbb{R}) \to \mathbb{R}$ so that the summary statistics $M^c, \mathcal{Q}, Q^c, T, G, P$(31) can be decomposed as

$$M^c = \int d\varrho \, m^c(\varrho), \tag{44}$$

$$Q^c = \int d\varrho \, q(\varrho)\varrho_c, \tag{45}$$

$$\mathcal{Q} = \int d\varrho \, q(\varrho), \tag{46}$$

$$T^{ck} = \int d\varrho \, \theta^{ck}(\varrho), \tag{47}$$

$$G = \int d\varrho \, g(\varrho), \tag{48}$$

$$P^c = \int d\varrho \, p^c(\varrho), \tag{49}$$

The following subsections focus on deriving the updates of the summary statistic densities $m(\cdot), q(\cdot), \theta(\cdot), g(\cdot)$ inherited from the SGD dynamics of the weight matrix $w$ (21).

### A.2.4 Overlap $m^k(\cdot)$

The expected increment for $m^k(\varrho)$ can also be decomposed as

$$\mathbb{E}[dm^k(\varrho)] = \mathbb{E}_t \mathbb{E}_c \left[ \mathbb{E}^c dm^{t,c}_k(\varrho) \right], \tag{50}$$

with

$$
\begin{aligned}
\frac{d}{2\eta}\mathbb{E}^c[dm_k^{t,c}(\varrho)_\gamma] = & -\sum_{\delta=1}^r I_{\sigma\sigma}^{t,c}(\gamma,\delta)m^k(\varrho)_\delta + (1-b\beta_t)I_\sigma^{t,c}(\gamma)\theta^{ck}(\varrho) \\
& + \frac{(1-b\beta_t)\beta_t\varrho_c m^k(\varrho)_\gamma - b\alpha_t^2 m^k(\varrho)_\gamma}{\Omega_{\gamma\gamma}^{t,c}}(I_{\sigma\omega}^{t,c}(\gamma,\gamma) - \beta_t M_\gamma^c I_\sigma^{t,c}(\gamma)) \\
& - \sum_{\delta\neq\gamma}\frac{(\alpha_t^2 + \beta_t^2\varrho_c)\mathcal{Q}_{\gamma\delta}}{\Omega_{\gamma\gamma}^{t,c}\Omega_{\delta\delta}^{t,c} - (\Omega_{\gamma\delta}^{t,c})^2}\Bigg[\big(I_{\sigma'\sigma\omega}^{t,c}(\gamma,\delta,\gamma) - \beta_t M_\gamma^c I_{\sigma'\sigma}^{t,c}(\gamma,\delta)\big)(\Omega_{\delta\delta}^{t,c}m^k(\varrho)_\gamma - \Omega_{\gamma\delta}^{t,c}m^k(\varrho)_\delta) \\
& \qquad\qquad\qquad + \big(I_{\sigma'\sigma\omega}^{t,c}(\gamma,\delta,\delta) - \beta_t M_\delta^c I_{\sigma'\sigma}^{t,c}(\gamma,\delta)\big)(\Omega_{\gamma\gamma}^{t,c}m^k(\varrho)_\delta - \Omega_{\gamma\delta}^{t,c}m^k(\varrho)_\gamma)\Bigg] \\
& - \frac{\mathcal{Q}_{\gamma\gamma}(\alpha_t^2 + \beta_t^2\varrho_c)m^k(\varrho)_\gamma}{\Omega_{\gamma\gamma}^{t,c}}(I_{\sigma'\sigma\omega}^{t,c}(\gamma,\gamma,\gamma) - \beta_t M_\gamma^c I_{\sigma'\sigma}^{t,c}(\gamma,\gamma)) - \beta_t\sum_{\delta=1}^r \mathcal{Q}_{\gamma\delta}I_{\sigma'\sigma}^{t,c}(\gamma,\delta)\theta^{ck}(\varrho) \\
& + \frac{\alpha_t^2(1-b\beta_t)Q_{\gamma\gamma}^c m^k(\varrho)_\gamma}{Q_{\gamma\gamma}^c\Omega_{\gamma\gamma}^{t,c} - \beta_t^2(Q_{\gamma\gamma}^c)^2}\Bigg[I_{\lambda^1\sigma'\omega}^{t,c}(\gamma,\gamma,\gamma) - \beta_t I_{(\lambda^1)^2\sigma'}^{t,c}(\gamma,\gamma)\Bigg] \\
& - \frac{\alpha_t^2 b m^k(\varrho)_\gamma}{\mathcal{Q}_{\gamma\gamma}\Omega_{\gamma\gamma}^{t,c} - \alpha_t^2(\mathcal{Q}_{\gamma\gamma})^2}\Bigg[I_{(\lambda^0)^2\sigma'}^{t,c}(\gamma,\gamma)(\Omega_{\gamma\gamma}^{t,c} - \alpha_t^2\mathcal{Q}_{\gamma\gamma})\Bigg] \\
& + \beta_t(1-b\beta_t)I_{\lambda^1\sigma'}^{t,c}(\gamma,\gamma)\theta^{ck}(\varrho) \\
& + \frac{\beta_t(1-b\beta_t)\varrho_c m^k(\varrho)_\gamma}{Q_{\gamma\gamma}^c\Omega_{\gamma\gamma}^{t,c} - \beta_t^2(Q_{\gamma\gamma}^c)^2}\Bigg[(I_{(\lambda^1)^2\sigma'}^{t,c}(\gamma,\gamma) - M_\gamma^c I_{\lambda^1\sigma'}^{t,c}(\gamma,\gamma))(\Omega_{\gamma\gamma}^{t,c} - \beta_t^2 Q_{\gamma\gamma}^c)\Bigg] \\
& - \alpha_t\beta_t b I_{\lambda^0\sigma'}(\gamma,\gamma)\theta^{ck}(\varrho) \\
& - \frac{\mathcal{Q}_{\gamma\gamma}b\beta_t\alpha_t^2\varrho_c m^k(\varrho)_\gamma}{\mathcal{Q}_{\gamma\gamma}\Omega_{\gamma\gamma}^{t,c} - \alpha_t^2(\mathcal{Q}_{\gamma\gamma})^2}\Bigg[I_{\lambda^0\sigma'\omega}^{t,c}(\gamma,\gamma,\gamma) - \alpha_t I_{(\lambda^0)^2\sigma'}^{t,c}(\gamma,\gamma) - M_\gamma^c\beta_t I_{\lambda^0\sigma'}^{t,c}(\gamma,\gamma)\Bigg] \\
& - \lambda m^k(\varrho)_\gamma
\end{aligned}
\tag{51}
$$

### A.2.5 Overlap $g(\cdot)$

The expected SGD for $g(\varrho)$ can be derived along nearly identical lines. By the same token, for $1 \le i \le R$, the decomposition

$$
\mathbb{E}[dg_i(\varrho)] = \mathbb{E}_t\mathbb{E}_c\left[\mathbb{E}^c dg_i^{t,c}(\varrho)\right]
\tag{52}
$$

holds with

$$
\frac{d}{2\eta}\mathbb{E}^c[dg^{t,c}(\varrho)_\gamma] = -\sum_{\delta=1}^r I_{\sigma\sigma}^{t,c}(\gamma,\delta)g_i(\varrho)_\delta + (1 - b\beta_t)I_\sigma^{t,c}(\gamma)p_i^c(\varrho)
$$

$$
+ \frac{(1 - b\beta_t)\beta_t\varrho_c g_i(\varrho)_\gamma - b\alpha_t^2 g_i(\varrho)_\gamma}{\Omega_{\gamma\gamma}^{t,c}}(I_{\sigma\omega}^{t,c}(\gamma,\gamma) - \beta_t M_\gamma^c I_\sigma^{t,c}(\gamma))
$$

$$
- \sum_{\delta\neq\gamma} \frac{(\alpha_t^2 + \beta_t^2\varrho_c)\mathcal{Q}_{\gamma\delta}}{\Omega_{\gamma\gamma}^{t,c}\Omega_{\delta\delta}^{t,c} - (\Omega_{\gamma\delta}^{t,c})^2}\left[\left(I_{\sigma'\sigma\omega}^{t,c}(\gamma,\delta,\gamma) - \beta_t M_\gamma^c I_{\sigma'\sigma}^{t,c}(\gamma,\delta)\right)(\Omega_{\delta\delta}^{t,c}g_i(\varrho)_\gamma - \Omega_{\gamma\delta}^{t,c}g_i(\varrho)_\delta)\right.
$$

$$
\left. + \left(I_{\sigma'\sigma\omega}^{t,c}(\gamma,\delta,\delta) - \beta_t M_\delta^c I_{\sigma'\sigma}^{t,c}(\gamma,\delta)\right)(\Omega_{\gamma\gamma}^{t,c}g(\varrho)_\delta - \Omega_{\gamma\delta}^{t,c}g(\varrho)_\gamma)\right]
$$

$$
- \frac{\mathcal{Q}_{\gamma\gamma}(\alpha_t^2 + \beta_t^2\varrho_c)g_i(\varrho)_\gamma}{\Omega_{\gamma\gamma}^{t,c}}(I_{\sigma'\sigma\omega}^{t,c}(\gamma,\gamma,\gamma) - \beta_t M_\gamma^c I_{\sigma'\sigma}^{t,c}(\gamma,\gamma)) - \beta_t\sum_{\delta=1}^r \mathcal{Q}_{\gamma\delta}I_{\sigma'\sigma}^{t,c}(\gamma,\delta)p_i^c(\varrho)
$$

$$
+ \frac{\alpha_t^2(1 - b\beta_t)Q_{\gamma\gamma}^c g_i(\varrho)_\gamma}{Q_{\gamma\gamma}^c\Omega_{\gamma\gamma}^{t,c} - \beta_t^2(Q_{\gamma\gamma}^c)^2}\left[I_{\lambda^1\sigma'\omega}^{t,c}(\gamma,\gamma,\gamma) - \beta_t I_{(\lambda^1)^2\sigma'}^{t,c}(\gamma,\gamma)\right]
$$

$$
- \frac{\alpha_t^2 b g_i(\varrho)_\gamma}{\mathcal{Q}_{\gamma\gamma}\Omega_{\gamma\gamma}^{t,c} - \alpha_t^2(\mathcal{Q}_{\gamma\gamma})^2}\left[I_{(\lambda^0)^2\sigma'}^{t,c}(\gamma,\gamma)(\Omega_{\gamma\gamma}^{t,c} - \alpha_t^2\mathcal{Q}_{\gamma\gamma})\right]
$$

$$
+ \beta_t(1 - b\beta_t)I_{\lambda^1\sigma'}^{t,c}(\gamma,\gamma)p_i^c(\varrho)
$$

$$
+ \frac{\beta_t(1 - b\beta_t)\varrho_c g_i(\varrho)_\gamma}{Q_{\gamma\gamma}^c\Omega_{\gamma\gamma}^{t,c} - \beta_t^2(Q_{\gamma\gamma}^c)^2}\left[(I_{(\lambda^1)^2\sigma'}^{t,c}(\gamma,\gamma) - M_\gamma^c I_{\lambda^1\sigma'}^{t,c}(\gamma,\gamma))(\Omega_{\gamma\gamma}^{t,c} - \beta_t^2 Q_{\gamma\gamma}^c)\right]
$$

$$
- \alpha_t\beta_t b I_{\lambda^0\sigma'}(\gamma,\gamma)p_i^c(\varrho)
$$

$$
- \frac{\mathcal{Q}_{\gamma\gamma}b\beta_t\alpha_t^2\varrho_c g_i(\varrho)_\gamma}{\mathcal{Q}_{\gamma\gamma}\Omega_{\gamma\gamma}^{t,c} - \alpha_t^2(\mathcal{Q}_{\gamma\gamma})^2}\left[I_{\lambda^0\sigma'\omega}^{t,c}(\gamma,\gamma,\gamma) - \alpha_t I_{(\lambda^0)^2\sigma'}^{t,c}(\gamma,\gamma) - M_\gamma^c\beta_t I_{\lambda^0\sigma'}^{t,c}(\gamma,\gamma)\right]
$$

$$
- \lambda g_i(\varrho)_\gamma, \tag{53}
$$

yielding the increment of $g(\cdot)$ under the SGD dynamics.

### A.2.6 Overlap $q(\cdot)$

We now turn to the summary statistic $q(\varrho)$ (31). First note that

$$
\mathbb{E}[d\mathcal{Q}] = \frac{1}{d}\sum_{i=1}^d \mathbb{E}_c\left[w_i\mathbb{E}_t[\mathbb{E}^c dw_i^{t,c}]^\top + \mathbb{E}_t[\mathbb{E}^c dw_i^{t,c}]w_i^\top + \mathbb{E}_{t,t'}[\mathbb{E}^c dw_i^{t,c}(dw_i^{t',c})^\top]\right]
$$

$$
\equiv \mathbb{E}_t\mathbb{E}_c[\mathbb{E}^c d\mathcal{Q}_{(1)}^{t,c}(\varrho)] + \mathbb{E}_{t,t'}\mathbb{E}_c[\mathbb{E}^c d\mathcal{Q}_{(2)}^{t,t',c}(\varrho)]. \tag{54}
$$

We have separated the linear term and the quadratic term. It follows that the density statistic $q(\cdot)$ can be similarly decomposed as

$$
\mathbb{E}[dq(\varrho)] = \mathbb{E}_t\mathbb{E}_c[\mathbb{E}^c dq_{(1)}^{t,c}(\varrho)] + \mathbb{E}_{t,t'}\mathbb{E}_c[\mathbb{E}^c dq_{(2)}^{t,t',c}(\varrho)]. \tag{55}
$$

In the following, we sequentially examine the linear and quadratic terms. The expected increment for the linear term $dq_{(1)}^{t,c}(\cdot)$ can be read from (21) as

$$\frac{d}{2\eta}\mathbb{E}^c[(dq_{(1)}^{t,c}(\varrho))_{\gamma\delta}] =$$

$$-\sum_{\epsilon=1}^r I_{\sigma\sigma}^{t,c}(\gamma,\epsilon)q(\varrho)_{\delta\epsilon} + (1-b\beta_t)I_\sigma^{t,c}(\gamma)m^c(\varrho)_\delta$$

$$+\frac{((1-b\beta_t)\beta_t\varrho_c - b\alpha_t^2)q(\varrho)_{\gamma\delta}}{\Omega_{\gamma\gamma}^{t,c}}(I_{\sigma\omega}^{t,c}(\gamma,\gamma) - \beta_t M_\gamma^c I_\sigma^{t,c}(\gamma))$$

$$-\sum_{\epsilon\neq\gamma}\frac{(\alpha_t^2 + \beta_t^2\varrho_c)\mathcal{Q}_{\epsilon\gamma}}{\Omega_{\gamma\gamma}^{t,c}\Omega_{\epsilon\epsilon}^{t,c} - (\Omega_{\gamma\epsilon}^{t,c})^2}\left[\left(I_{\sigma'\sigma\omega}^{t,c}(\gamma,\epsilon,\gamma) - \beta_t M_\gamma^c I_{\sigma'\sigma}^{t,c}(\gamma,\epsilon)\right)(\Omega_{\epsilon\epsilon}^{t,c}q(\varrho)_{\gamma\delta} - \Omega_{\gamma\epsilon}^{t,c}q(\varrho)_{\epsilon\delta})\right.$$

$$\left.+\left(I_{\sigma'\sigma\omega}^{t,c}(\gamma,\epsilon,\epsilon) - \beta_t M_\epsilon^c I_{\sigma'\sigma}^{t,c}(\gamma,\epsilon)\right)(\Omega_{\gamma\gamma}^{t,c}q(\varrho)_{\delta\epsilon} - \Omega_{\gamma\epsilon}^{t,c}q(\varrho)_{\gamma\delta})\right]$$

$$-\frac{(\alpha_t^2 + \beta_t^2\varrho_c)\mathcal{Q}_{\gamma\gamma}q(\varrho)_{\gamma\delta}}{\Omega_{\gamma\gamma}^{t,c}}(I_{\sigma'\sigma\omega}^{t,c}(\gamma,\gamma,\gamma) - \beta_t M_\gamma^c I_{\sigma'\sigma}^{t,c}(\gamma,\gamma)) - \beta_t\sum_{\epsilon=1}^r \mathcal{Q}_{\epsilon\gamma}I_{\sigma'\sigma}^{t,c}(\gamma,\epsilon)m^c(\varrho)_\delta$$

$$+\frac{Q_{\gamma\gamma}^c\alpha_t^2(1-b\beta_t)q(\varrho)_{\gamma\delta}}{Q_{\gamma\gamma}^c\Omega_{\gamma\gamma}^{t,c} - \beta_t^2(Q_{\gamma\gamma}^c)^2}\left[I_{\lambda^1\sigma'\omega}^{t,c}(\gamma,\gamma,\gamma) - \beta_t I_{(\lambda^1)^2\sigma'}^{t,c}(\gamma,\gamma)\right]$$

$$-\frac{\alpha_t^2 bq(\varrho)_{\gamma\delta}}{\mathcal{Q}_{\gamma\gamma}\Omega_{\gamma\gamma}^{t,c} - \alpha_t^2(\mathcal{Q}_{\gamma\gamma})^2}\left[I_{(\lambda^0)^2\sigma'}^{t,c}(\gamma,\gamma)(\Omega_{\gamma\gamma}^{t,c} - \alpha_t^2\mathcal{Q}_{\gamma\gamma})\right] + \beta_t(1-b\beta_t)I_{\lambda^1\sigma'}^{t,c}(\gamma,\gamma)m^c(\varrho)_\delta$$

$$+\frac{\beta_t(1-b\beta_t)q(\varrho)_{\gamma\delta}\varrho^c}{Q_{\gamma\gamma}^c\Omega_{\gamma\gamma}^{t,c} - \beta_t^2(Q_{\gamma\gamma}^c)^2}\left[(I_{(\lambda^1)^2\sigma'}^{t,c}(\gamma,\gamma) - M_\gamma^c I_{\lambda^1\sigma'}^{t,c}(\gamma,\gamma)))(\Omega_{\gamma\gamma}^{t,c} - \beta_t^2 Q_{\gamma\gamma}^c)\right] - \alpha_t\beta_t b I_{\lambda^0\sigma'}^{t,c}(\gamma,\gamma)m^c(\varrho)_\delta$$

$$-\frac{\alpha_t^2\beta_t b\mathcal{Q}_{\gamma\gamma}\varrho^c q(\varrho)_{\gamma\delta}}{\mathcal{Q}_{\gamma\gamma}\Omega_{\gamma\gamma}^{t,c} - \alpha_t^2(\mathcal{Q}_{\gamma\gamma})^2}\left[I_{\lambda^0\sigma'\omega}^{t,c}(\gamma,\gamma,\gamma) - \alpha_t I_{(\lambda^0)^2\sigma'}^{t,c}(\gamma,\gamma) - M_\gamma^c\beta_t I_{\lambda^0\sigma'}^{t,c}(\gamma,\gamma)\right]$$

$$-\lambda q(\varrho)_{\gamma\delta}$$

$$+(\gamma\leftrightarrow\delta) \tag{56}$$

We now turn to the quadratic term $dq_{(2)}^{t,c}(\cdot)$. Keeping only leading order terms,

$$\frac{d}{4\eta^2}\mathbb{E}^c[(dq_{(2)}^{t,c}(\varrho))_{\gamma\delta}] = \frac{1}{2}\nu(\varrho)I_{\sigma\sigma}^{t,t',c}(\gamma,\delta)\left[(1-b\beta_t)(1-b\beta_{t'})\varrho_c + b^2\alpha_t\alpha_{t'}\right]$$

$$-\nu(\varrho)\sum_{\epsilon=1}^r I_{\sigma\sigma\sigma'}^{t,t,t',c}(\gamma,\epsilon,\delta)\mathcal{Q}_{\epsilon\delta}\left[\beta_{t'}(1-b\beta_t)\varrho_c - b\alpha_t\alpha_{t'}\right]$$

$$+\nu(\varrho)\left((1-b\beta_{t'})I_{\sigma\sigma'\lambda^1}^{t,t',c}(\gamma,\delta,\delta) - b\alpha_{t'}I_{\sigma\sigma'\lambda^0}^{t,t',c}(\gamma,\delta,\delta)\right)\left[\beta_{t'}(1-b\beta_t)\varrho_c - b\alpha_t\alpha_{t'}\right]$$

$$+\frac{1}{2}\nu(\varrho)\sum_{\epsilon,\iota}I_{\sigma'\sigma\sigma'\sigma}^{t,t,t',t',c}(\gamma,\epsilon,\delta,\iota)\mathcal{Q}_{\gamma\epsilon}\mathcal{Q}_{\delta\iota}\left[\beta_t\beta_{t'}\varrho_c + \alpha_t\alpha_{t'}\right]$$

$$-\nu(\varrho)\sum_{\epsilon=1}^r\left((1-b\beta_{t'})I_{\sigma'\sigma\sigma'\lambda^1}^{t,t,t',c}(\gamma,\epsilon,\delta,\delta) - b\alpha_{t'}I_{\sigma'\sigma\sigma'\lambda^0}^{t,t,t',c}(\gamma,\epsilon,\delta,\delta)\right)\mathcal{Q}_{\epsilon\gamma}\left[\beta_t\beta_{t'}\varrho_c + \alpha_t\alpha_{t'}\right]$$

$$+\frac{1}{2}\left((1-b\beta_{t'})(1-b\beta_t)I_{\sigma'\sigma'\lambda^1\lambda^1}^{t,t',t,t',c}(\gamma,\delta,\gamma,\delta) - (1-b\beta_{t'})b\alpha_t I_{\sigma'\sigma'\lambda^0\lambda^1}^{t,t',t,t',c}(\gamma,\delta,\gamma,\delta))\right.$$

$$\left.-b\alpha_{t'}(1-b\beta_t)I_{\sigma'\sigma'\lambda^0\lambda^1}^{t,t',t,t,c}(\gamma,\delta,\delta,\gamma) + b^2\alpha_t\alpha_{t'}I_{\sigma'\sigma'\lambda^0\lambda^0}^{t,t',t,t',c}(\gamma,\delta,\gamma,\delta)\right)\nu(\varrho)(\beta_t\beta_{t'}\varrho_c + \alpha_t\alpha_{t'})$$

$$+(\gamma\leftrightarrow\delta) \tag{57}$$

We introduced the integrals

$$I_{\sigma\sigma}^{t,t',c}(\gamma,\delta) = \mathbb{E}_{\omega_\gamma,\omega_\delta}[\sigma(\omega_\gamma + v_\gamma p_t)\sigma(\omega_\delta + v_\delta p_t)]$$

$$\omega_\gamma,\omega_\delta \sim \mathcal{N}\left((\beta_t,\beta_{t'})\odot M_{(\gamma,\delta)}^c, \Omega_{(\gamma,\delta)}^{t,t',c}\right) \tag{58}$$

$$I_{\sigma\sigma\sigma'}^{t_1,t_2,t_3,c}(\gamma,\epsilon,\delta) = \mathbb{E}_{\omega_\gamma,\omega_\epsilon,\omega_\delta}[\sigma(\omega_\gamma + v_\gamma p_t)\sigma(\omega_\epsilon + v_\epsilon p_t)\sigma'(\omega_\delta + v_\delta p_t)],$$

$$\omega_\gamma,\omega_\epsilon,\omega_\delta \sim \mathcal{N}\left((\beta_{t_1},\beta_{t_2},\beta_{t_3})\odot M_{(\gamma,\epsilon,\delta)}^c, \Omega_{(\gamma,\epsilon,\delta)}^{(3),t_1,t_2,t_3,c}\right) \tag{59}$$

$$I_{\sigma\sigma',\lambda^1}^{t_1,t_2,t_3,c}(\gamma,\epsilon,\delta) = \mathbb{E}_{\omega_\gamma,\omega_\epsilon,\lambda_\delta^1}[\sigma(\omega_\gamma + v_\gamma p_t)\sigma'(\omega_\epsilon + v_\epsilon p_t)\lambda_\delta^1],$$

$$\omega_\gamma,\omega_\epsilon,\lambda_\delta^1 \sim \mathcal{N}\left((\beta_{t_1},\beta_{t_2},1)\odot M_{(\gamma,\epsilon,\delta)}^c, \Phi_{(\gamma,\epsilon,\delta)}^{(3),t_1,t_2,t_3,c}\right) \tag{60}$$

$$I_{\sigma\sigma',\lambda^0}^{t_1,t_2,t_3,c}(\gamma,\epsilon,\delta) = \mathbb{E}_{\omega_\gamma,\omega_\epsilon,\lambda_\delta^0}[\sigma(\omega_\gamma + v_\gamma p_t)\sigma'(\omega_\epsilon + v_\epsilon p_t)\lambda_\delta^0],$$

$$\omega_\gamma,\omega_\epsilon,\lambda_\delta^0 \sim \mathcal{N}\left((\beta_{t_1},\beta_{t_2},0)\odot M_{(\gamma,\epsilon,\delta)}^c, \Psi_{(\gamma,\epsilon,\delta)}^{(3),t_1,t_2,t_3,c}\right) \tag{61}$$

$$I_{\sigma'\sigma\sigma'\sigma}^{t_1,t_2,t_3,t_4,c}(\gamma,\epsilon,\delta,\iota) = \mathbb{E}_{\omega_\gamma,\omega_\epsilon,\omega_\delta,\omega_\iota}[\sigma'(\omega_\gamma + v_\gamma p_t)\sigma(\omega_\epsilon + v_\epsilon p_t)\sigma'(\omega_\delta)\sigma(\omega_\iota + v_\iota p_t)] \quad,$$

$$\omega_\gamma,\omega_\epsilon,\omega_\delta,\omega_\iota, \sim \mathcal{N}\left((\beta_{t_1},\beta_{t_2},\beta_{t_3},\beta_{t_4})\odot M_{(\gamma,\epsilon,\delta,\iota)}^c, \Omega_{(\gamma,\epsilon,\delta,\iota)}^{(4),t_1,t_2,t_3,t_4,c}\right) \tag{62}$$

$$I_{\sigma'\sigma\sigma'\lambda^1}^{t_1,t_2,t_3,t_4,c}(\gamma,\epsilon,\delta,\iota) = \mathbb{E}_{\omega_\gamma,\omega_\epsilon,\omega_\delta,\lambda_\iota^1}[\sigma'(\omega_\gamma + v_\gamma p_t)\sigma(\omega_\epsilon + v_\epsilon p_t)\sigma'(\omega_\delta + v_\delta p_t)\lambda_\iota^1] \quad,$$

$$\omega_\gamma,\omega_\epsilon,\omega_\delta,\lambda_\iota^1, \sim \mathcal{N}\left((\beta_{t_1},\beta_{t_2},\beta_{t_3},1)\odot M_{(\gamma,\epsilon,\delta,\iota)}^c, \Phi_{(\gamma,\epsilon,\delta,\iota)}^{(4),t_1,t_2,t_3,t_4,c}\right) \tag{63}$$

$$I_{\sigma'\sigma\sigma'\lambda^0}^{t_1,t_2,t_3,t_4,c}(\gamma,\epsilon,\delta,\iota) = \mathbb{E}_{\omega_\gamma,\omega_\epsilon,\omega_\delta,\lambda_\iota^0}[\sigma'(\omega_\gamma + v_\gamma p_t)\sigma(\omega_\epsilon + v_\epsilon p_t)\sigma'(\omega_\delta + v_\delta p_t)\lambda_\iota^0] \quad,$$

$$\omega_\gamma,\omega_\epsilon,\omega_\delta,\lambda_\iota^0, \sim \mathcal{N}\left((\beta_{t_1},\beta_{t_2},\beta_{t_3},0)\odot M_{(\gamma,\epsilon,\delta,\iota)}^c, \Psi_{(\gamma,\epsilon,\delta,\iota)}^{(4),t_1,t_2,t_3,t_4,c}\right) \tag{64}$$

$$I_{\sigma'\sigma'\lambda^1\lambda^1}^{t_1,t_2,t_3,t_4,c}(\gamma,\epsilon,\delta,\iota) = \mathbb{E}_{\omega_\gamma,\omega_\epsilon,\lambda_\delta^1,\lambda_\iota^1}[\sigma'(\omega_\gamma + v_\gamma p_t)\sigma'(\omega_\epsilon + v_\epsilon p_t)\lambda_\delta^1\lambda_\iota^1] \quad,$$

$$\omega_\gamma,\omega_\epsilon,\lambda_\delta^1,\lambda_\iota^1 \sim \mathcal{N}\left((\beta_{t_1},\beta_{t_2},1,1)\odot M_{(\gamma,\epsilon,\delta,\iota)}^c, P_{(\gamma,\epsilon,\delta,\iota)}^{(4,1,1),t_1,t_2,t_3,t_4,c}\right) \tag{65}$$

$$I_{\sigma'\sigma'\lambda^0\lambda^1}^{t_1,t_2,t_3,t_4,c}(\gamma,\epsilon,\delta,\iota) = \mathbb{E}_{\omega_\gamma,\omega_\epsilon,\lambda_\delta^0,\lambda_\iota^1}[\sigma'(\omega_\gamma + v_\gamma p_t)\sigma'(\omega_\epsilon + v_\epsilon p_t)\lambda_\delta^0\lambda_\iota^1] \quad,$$

$$\omega_\gamma,\omega_\epsilon,\lambda_\delta^0,\lambda_\iota^1 \sim \mathcal{N}\left((\beta_{t_1},\beta_{t_2},0,1)\odot M_{(\gamma,\epsilon,\delta,\iota)}^c, P_{(\gamma,\epsilon,\delta,\iota)}^{(4,0,1),t_1,t_2,t_3,t_4,c}\right) \tag{66}$$

$$I_{\sigma'\sigma'\lambda^0\lambda^0}^{t_1,t_2,t_3,t_4,c}(\gamma,\epsilon,\delta,\iota) = \mathbb{E}_{\omega_\gamma,\omega_\epsilon,\lambda_\delta^0,\lambda_\iota^0}[\sigma'(\omega_\gamma + v_\gamma p_t)\sigma'(\omega_\epsilon + v_\epsilon p_t)\lambda_\delta^0\lambda_\iota^0] \quad,$$

$$\omega_\gamma,\omega_\epsilon,\lambda_\delta^0,\lambda_\iota^1 \sim \mathcal{N}\left((\beta_{t_1},\beta_{t_2},0,0)\odot M_{(\gamma,\epsilon,\delta,\iota)}^c, P_{(\gamma,\epsilon,\delta,\iota)}^{(4,0,0),t_1,t_2,t_3,t_4,c}\right). \tag{67}$$

We further denoted

$$\Omega^{t,t',c} = \alpha_t \alpha_{t'} \mathcal{Q} + \beta_t \beta_{t'} Q^c$$

$$\Omega^{(3),t_1,t_2,t_3,c}_{(\gamma,\epsilon,\delta)} = \begin{pmatrix} \Omega^{t_1,t_1,c}_{\gamma\gamma} & \Omega^{t_1,t_2,c}_{\gamma\epsilon} & \Omega^{t_1,t_3,c}_{\gamma\delta} \\ \Omega^{t_2,t_1,c}_{\epsilon\gamma} & \Omega^{t_2,t_2,c}_{\epsilon\epsilon} & \Omega^{t_2,t_3,c}_{\epsilon\delta} \\ \Omega^{t_3,t_1,c}_{\delta\gamma} & \Omega^{t_3,t_2,c}_{\delta\epsilon} & \Omega^{t_3,t_3,c}_{\delta\delta} \end{pmatrix}$$

$$\Phi^{(3),t_1,t_2,t_3,c}_{(\gamma,\epsilon,\delta)} = \begin{pmatrix} \Omega^{t_1,t_1,c}_{\gamma\gamma} & \Omega^{t_1,t_2,c}_{\gamma\epsilon} & \beta_{t_1} Q^c_{\gamma\delta} \\ \Omega^{t_2,t_1,c}_{\epsilon\gamma} & \Omega^{t_2,t_2,c}_{\epsilon\epsilon} & \beta_{t_2} Q^c_{\epsilon\delta} \\ \beta_{t_1} Q^c_{\gamma\delta} & \beta_{t_2} Q^c_{\epsilon\delta} & Q^c_{\delta\delta} \end{pmatrix}$$

$$\Psi^{(3),t_1,t_2,t_3,c}_{(\gamma,\epsilon,\delta)} = \begin{pmatrix} \Omega^{t_1,t_1,c}_{\gamma\gamma} & \Omega^{t_1,t_2,c}_{\gamma\epsilon} & \alpha_{t_1} \mathcal{Q}_{\gamma\delta} \\ \Omega^{t_2,t_1,c}_{\epsilon\gamma} & \Omega^{t_2,t_2,c}_{\epsilon\epsilon} & \alpha_{t_2} \mathcal{Q}_{\epsilon\delta} \\ \alpha_{t_1} Q^c_{\gamma\delta} & \alpha_{t_2} \mathcal{Q}_{\epsilon\delta} & \mathcal{Q}_{\delta\delta} \end{pmatrix}$$

$$\Omega^{(4),t_1,t_2,t_3,t_4,c}_{(\gamma,\epsilon,\delta,\iota)} = \begin{pmatrix} \Omega^{t_1,t_1,c}_{\gamma\gamma} & \Omega^{t_1,t_2,c}_{\gamma\epsilon} & \Omega^{t_1,t_3,c}_{\gamma\delta} & \Omega^{t_1,t_4,c}_{\gamma\iota} \\ \Omega^{t_2,t_1,c}_{\epsilon\gamma} & \Omega^{t_2,t_2,c}_{\epsilon\epsilon} & \Omega^{t_2,t_3,c}_{\epsilon\delta} & \Omega^{t_2,t_4,c}_{\epsilon\iota} \\ \Omega^{t_3,t_1,c}_{\delta\gamma} & \Omega^{t_3,t_2,c}_{\delta\epsilon} & \Omega^{t_3,t_3,c}_{\delta\delta} & \Omega^{t_3,t_4,c}_{\delta\iota} \\ \Omega^{t_4,t_1,c}_{\gamma\iota} & \Omega^{t_4,t_2,c}_{\epsilon\iota} & \Omega^{t_4,t_3,c}_{\delta\iota} & \Omega^{t_4,t_4,c}_{\iota\iota} \end{pmatrix}$$

$$\Phi^{(4),t_1,t_2,t_3,t_4,c}_{(\gamma,\epsilon,\delta,\iota)} = \begin{pmatrix} \Omega^{t_1,t_1,c}_{\gamma\gamma} & \Omega^{t_1,t_2,c}_{\gamma\epsilon} & \Omega^{t_1,t_3,c}_{\gamma\delta} & \beta_{t_1} Q^c_{\gamma\iota} \\ \Omega^{t_2,t_1,c}_{\epsilon\gamma} & \Omega^{t_2,t_2,c}_{\epsilon\epsilon} & \Omega^{t_2,t_3,c}_{\epsilon\delta} & \beta_{t_2} Q^c_{\epsilon\iota} \\ \Omega^{t_3,t_1,c}_{\delta\gamma} & \Omega^{t_3,t_2,c}_{\delta\epsilon} & \Omega^{t_3,t_3,c}_{\delta\delta} & \beta_{t_3} Q^c_{\delta\iota} \\ \beta_{t_1} Q^c_{\gamma\iota} & \beta_{t_2} Q^c_{\epsilon\iota} & \beta_{t_3} Q^c_{\delta\iota} & Q^c_{\iota\iota} \end{pmatrix}$$

$$\Psi^{(4),t_1,t_2,t_3,t_4,c}_{(\gamma,\epsilon,\delta,\iota)} = \begin{pmatrix} \Omega^{t_1,t_1,c}_{\gamma\gamma} & \Omega^{t_1,t_2,c}_{\gamma\epsilon} & \Omega^{t_1,t_3,c}_{\gamma\delta} & \alpha_{t_1} \mathcal{Q}_{\gamma\iota} \\ \Omega^{t_2,t_1,c}_{\epsilon\gamma} & \Omega^{t_2,t_2,c}_{\epsilon\epsilon} & \Omega^{t_2,t_3,c}_{\epsilon\delta} & \alpha_{t_2} \mathcal{Q}_{\epsilon\iota} \\ \Omega^{t_3,t_1,c}_{\delta\gamma} & \Omega^{t_3,t_2,c}_{\delta\epsilon} & \Omega^{t_3,t_3,c}_{\delta\delta} & \alpha_{t_3} \mathcal{Q}_{\delta\iota} \\ \alpha_{t_1} \mathcal{Q}_{\gamma\iota} & \alpha_{t_2} \mathcal{Q}_{\epsilon\iota} & \alpha_{t_3} \mathcal{Q}_{\delta\iota} & \mathcal{Q}_{\iota\iota} \end{pmatrix}$$

$$P^{(4,1,1),t_1,t_2,t_3,t_4,c}_{(\gamma,\epsilon,\delta,\iota)} = \begin{pmatrix} \Omega^{t_1,t_1,c}_{\gamma\gamma} & \Omega^{t_1,t_2,c}_{\gamma\epsilon} & \beta_{t_1} Q^c_{\gamma\delta} & \beta_{t_1} Q^c_{\gamma\iota} \\ \Omega^{t_2,t_1,c}_{\epsilon\gamma} & \Omega^{t_2,t_2,c}_{\epsilon\epsilon} & \beta_{t_2} Q^c_{\epsilon\delta} & \beta_{t_2} Q^c_{\epsilon\iota} \\ \beta_{t_1} Q^c_{\gamma\delta} & \beta_{t_2} Q^c_{\epsilon\delta} & Q^c_{\delta\delta} & Q^c_{\delta\iota} \\ \beta_{t_1} Q^c_{\gamma\iota} & \beta_{t_2} Q^c_{\epsilon\iota} & Q^c_{\iota\delta} & Q^c_{\iota\iota} \end{pmatrix}$$

$$P^{(4,0,1),t_1,t_2,t_3,t_4,c}_{(\gamma,\epsilon,\delta,\iota)} = \begin{pmatrix} \Omega^{t_1,t_1,c}_{\gamma\gamma} & \Omega^{t_1,t_2,c}_{\gamma\epsilon} & \alpha_{t_1} \mathcal{Q}_{\gamma\delta} & \beta_{t_1} Q^c_{\gamma\iota} \\ \Omega^{t_2,t_1,c}_{\epsilon\gamma} & \Omega^{t_2,t_2,c}_{\epsilon\epsilon} & \alpha_{t_2} \mathcal{Q}_{\epsilon\delta} & \beta_{t_2} Q^c_{\epsilon\iota} \\ \alpha_{t_1} \mathcal{Q}_{\gamma\delta} & \alpha_{t_2} \mathcal{Q}_{\epsilon\delta} & \mathcal{Q}_{\delta\delta} & 0 \\ \beta_{t_1} Q^c_{\gamma\iota} & \beta_{t_2} Q^c_{\epsilon\iota} & 0 & Q^c_{\iota\iota} \end{pmatrix}$$

$$P^{(4,0,0),t_1,t_2,t_3,t_4,c}_{(\gamma,\epsilon,\delta,\iota)} = \begin{pmatrix} \Omega^{t_1,t_1,c}_{\gamma\gamma} & \Omega^{t_1,t_2,c}_{\gamma\epsilon} & \alpha_{t_1} \mathcal{Q}_{\gamma\delta} & \alpha_{t_1} \mathcal{Q}_{\gamma\iota} \\ \Omega^{t_2,t_1,c}_{\epsilon\gamma} & \Omega^{t_2,t_2,c}_{\epsilon\epsilon} & \alpha_{t_2} \mathcal{Q}_{\epsilon\delta} & \alpha_{t_2} \mathcal{Q}_{\epsilon\iota} \\ \alpha_{t_1} \mathcal{Q}_{\gamma\delta} & \alpha_{t_2} \mathcal{Q}_{\epsilon\delta} & \mathcal{Q}_{\delta\delta} & \mathcal{Q}_{\delta\iota} \\ \alpha_{t_1} \mathcal{Q}_{\gamma\iota} & \alpha_{t_2} \mathcal{Q}_{\epsilon\iota} & \mathcal{Q}_{\iota\delta} & \mathcal{Q}_{\iota\iota} \end{pmatrix} \tag{68}$$

## A.3  Update for $v$

Finally, we can ascertain the asymptotic evolution of the time encoding weights $v$. In the considered limit, the update $dv$ again concentrates. As above, let us decompose the expected increment as

$$\mathbb{E}[dv_\gamma] = \mathbb{E}_t \mathbb{E}_c [\mathbb{E}^c dv^{t,c}_\gamma], \tag{69}$$

with

$$\mathbb{E}^c dv^{t,c}_\gamma = -\frac{2\eta}{d} \left[ \sum_\delta \mathcal{Q}_{\gamma\delta} I^{t,c}_{\sigma'\sigma}(\gamma,\delta) - (1 - b\beta_t) I^{t,c}_{\lambda^1\sigma'}(\gamma,\gamma) + b\alpha_t I^{t,c}_{\lambda^0\sigma'}(\gamma,\gamma) + \lambda v_\gamma \right] \tag{70}$$

## A.4 Continuous time limit

Equations (51),(53),(56) and (57) provide the update equations for the summary statistic densities $m(\cdot), g(\cdot), q(\cdot)$ under SGD steps (4), which take the form

$$\frac{d}{2\eta}dm(\varrho) = F_m(\varrho, m(\varrho), q(\varrho), M, Q, \mathcal{Q}, b),$$

$$\frac{d}{2\eta}dg = F_g(\varrho, m(\varrho), q(\varrho), M, Q, \mathcal{Q}, b),$$

$$\frac{d}{2\eta}dq = F_q(\varrho, m(\varrho), q(\varrho), M, Q, \mathcal{Q}, b), \tag{71}$$

where the update functions $F_{m,q,g}$ denote the right hand sides of (51),(53),(56) and (57), and we have omitted the time step indices to ease the notations. From (44), these updates translate directly at the level of the summary statistics $M, Q, \mathcal{Q}, G$ into

$$\frac{d}{2\eta}dM^c = F_M(M, Q, \mathcal{Q}, b, v)^c, \qquad F_M(\cdot)^c = \int F_m(\varrho, m(\varrho), q(\varrho), \cdot)^c d\varrho,$$

$$\frac{d}{2\eta}dG = F_G(M, Q, \mathcal{Q}, b, v), \qquad F_G(\cdot) = \int F_g(\varrho, m(\varrho), q(\varrho), \cdot) d\varrho,$$

$$\frac{d}{2\eta}dQ^c = F_Q(M, Q, \mathcal{Q}, b, v)^c, \qquad F_Q(\cdot)^c = \int \varrho_c F_q(\varrho, m(\varrho), q(\varrho), \cdot) d\varrho,$$

$$\frac{d}{2\eta}d\mathcal{Q} = F_{\mathcal{Q}}(M, Q, \mathcal{Q}, b, v), \qquad F_{\mathcal{Q}}(\cdot) = \int F_q(\varrho, m(\varrho), q(\varrho), \cdot) d\varrho. \tag{72}$$

We remind that from (17), the skip connection strength similarly obeys

$$\frac{d}{2\eta}db = F_b(b), \tag{73}$$

where the update function $F_b$ corresponds to the right hand side of equation (17). Similarly, from (70), the time encoding weights obey

$$\frac{d}{2\eta}dv = F_v(M, Q, \mathcal{Q}, b, v), \tag{74}$$

where $F_v$ corresponds to the right-hand side of (70). Now remark that in the asymptotic limit $d \to \infty$, the coefficient $d/2\eta$ tends to zero. Introducing the time variable $\vartheta \equiv 2\eta\mu/d$, so that $d\vartheta = 2\eta/d$, the discrete processes (72) and (73) are thus asymptotically described by the limiting ODEs

$$\frac{dM}{d\vartheta} = F_M(M, Q, \mathcal{Q}, b, v),$$

$$\frac{dQ}{d\vartheta} = F_Q(M, Q, \mathcal{Q}, b, v),$$

$$\frac{d\mathcal{Q}}{d\vartheta} = F_{\mathcal{Q}}(M, Q, \mathcal{Q}, b, v),$$

$$\frac{db}{d\vartheta} = F_b(b),$$

$$\frac{dv}{d\vartheta} = F_v(M, Q, \mathcal{Q}, b, v) \tag{75}$$

Finally, the ODE for $b$, governing the dynamics of the skip connection strength $b$ over the SGD optimization dynamics, can be solved in closed-form as

$$b(\vartheta) = \frac{\Lambda\mathbb{E}_t[\beta_t]}{\Lambda\mathbb{E}_t[\beta_t^2] + \mathbb{E}_t[\alpha_t^2]}\left[1 - e^{-\left(\Lambda\mathbb{E}_t[\beta_t^2] + \mathbb{E}_t[\alpha_t^2]\right)\vartheta}\right] + b_0 e^{-\left(\Lambda\mathbb{E}_t[\beta_t^2] + \mathbb{E}_t[\alpha_t^2]\right)\vartheta}, \tag{76}$$

where $b_0$ designates the value of $b$ at initialization. This completes the derivation of Result 2.1. $\square$

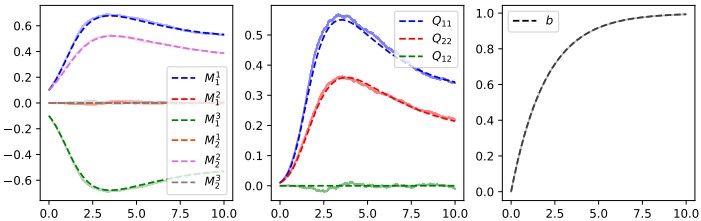

Figure 5: Evolution of the summary statistics (31) $M$ (**left**), $Q$ (**middle**) and skip connection strength $b$ (**right**), characterizing the dynamics of the AE parameters (3) under SGD dynamics (4). Parameters $\sigma = \tanh, r = 2, \lambda = 0, \eta = 0.2, \mathcal{G} = \{1/2\}$ were used, and the target density $\rho$ was taken to be a Gaussian mixture with three isotropic clusters (see also Fig. 7 in the main text). The weight vectors were initialized along the centroids of the target density, with norm 0.1, while the initial skip connection strength is $b_0 = 0$. Dashed lines: theoretical characterization of Result 2.1. Continuous lines: numerical experiments in $d = 1000$, for a single run.

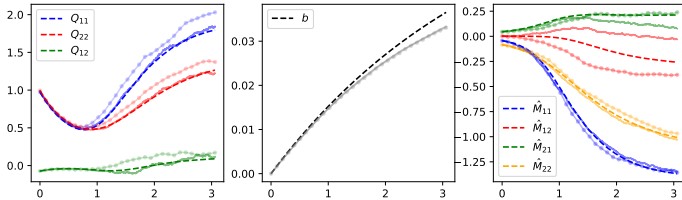

Figure 6: Evolution of the summary statistics (31) $Q$ (**left**), $b$ (**middle**) and skip connection strength $M$ (**right**), characterizing the dynamics of the AE parameters (3) under SGD dynamics (4). Parameters $\sigma = \tanh, r = 2, \lambda = 0.784, \eta = 0.2, \mathcal{G} = \{1/2\}$ were used; weights were initialized with random independent Gaussian components and $b_0 = 0$. Dotted continuous lines : numerical experiments for a target density $\rho$ given by the set of MNIST sevens. Continuous lines: numerical experiments for a unimodal Gaussian target distribution with covariance matching that of the set of MNIST sevens. Dashed lines: theoretical predictions of Result 2.1 for the latter Gaussian target density.

## A.5 Numerical validation

We plot the theoretical predictions of Result 2.1 for the evolution of the summary statistics $M, Q, \mathcal{Q}, G, b$ (31) under the SGD dynamics (4) in Fig. 5 for a Gaussian mixture target density $\rho$ with three isotropic modes, learnt by an AE with $r = 2$ hidden units and $\tanh$ activation, using learning rate $\eta = 0.2$ and weight decay $\lambda = 0$. The centroids of the clusters were taken as $\pm e_1, e_2$ for two orthonormal vectors $e_1, e_2$, and the columns of the weight matrix $w$ were initialized with a warm start as $0.1 \times e_{1,2}$. Finally, for simplicity, the expectation $\mathbb{E}_t$ in (4) was chosen to bear over a delta distribution around $\mathcal{G} = \{1/2\}$, instead of the full integral over $[0, 1]$. Including more points in the grid $\mathcal{G}$ was not found to significantly alter the qualitative aspect of the generated density. Fig. 5 reveals an overall good agreement between the theoretical predictions of Result 2.1 (dashed lines) and numerical experiments (solid lines), obtained by simulating the model in large but finite dimension $d = 1000$.

Fig. 6 similarly contrasts numerical experiments for a target distribution corresponding to MNIST images of sevens (dotted lines), a Gaussian target density with matching covariance (solid lines), and the theoretical predictions of Result 2.1 for the latter. All experimental details are specified in the caption. Although the agreement between the three curves is overall good, discrepancies appear, in particular due to the rather low dimensionality $d = 784$.

## A.6 Extensions

We briefly describe, for completeness, how the analysis can be generalized to characterize the learning of more complex DAE architectures. Namely, we discuss how the derivation can be adapted to accommodate (a) untied weights and (b) time encodings.

**Untying the weights –** The analysis reported in the present appendix can be extended to untied DAE architectures of the form

$$f_{b,u,v}(x) = b \times x + \frac{u}{\sqrt{d}} \sigma \left( \frac{v^\top x}{\sqrt{d}} \right), \tag{77}$$

trained with online SGD

$$b_{\mu+1} - b_\mu = - \frac{\eta}{d^2} \left( \partial_b \mathbb{E}_t \left\| x_1^\mu - f_{b_\mu, u_\mu, v_\mu} \left( \alpha_t x_0^\mu + \beta_t x_1^\mu \right) \right\|^2 \right), \tag{78}$$

$$u_{\mu+1} - u_\mu = - \eta \nabla_u \mathbb{E}_t \left\| x_1^\mu - f_{b_\mu, u_\mu, v_\mu} \left( \alpha_t x_0^\mu + \beta_t x_1^\mu \right) \right\|^2 - \eta \frac{\lambda}{d} u_\mu, \tag{79}$$

$$v_{\mu+1} - v_\mu = - \eta \nabla_v \mathbb{E}_t \left\| x_1^\mu - f_{b_\mu, u_\mu, v_\mu} \left( \alpha_t x_0^\mu + \beta_t x_1^\mu \right) \right\|^2 - \eta \frac{\lambda}{d} v_\mu. \tag{80}$$

Such an extension, however, comes at the price of more cumbersome expressions, as the summary statistics (31) needs to be introduced for the two sets of weights $u, v$, in addition to cross-statistics of the form $u^\top \Sigma(c) v/d$ and $u^\top v/d$. We refer the interested reader to Appendix B of [56] where such a derivation is detailed, in a closely related setting. Experimentally, in the probed settings, we did not observe a significant effect of (un)tying the weights on the qualitative phenomenology discussed in the main text.

**Deeper architectures –** This manuscript focuses on the simpler instance of DAE architecture (3), namely a two-layer model with a skip connection. It is of interest to remark, however, that the technical ideas employed in its analysis can be extended to study more intricate models, at the cost of considerably heavier mathematical expression. For this reason, we present in this paragraph a model of multi-layer DAE which we anticipate is amenable to similar analysis, and sketch the main ideas underlying the latter; the detailed analysis, however, is left for future work.

Consider the depth-$2L$ model $f_{\{b_\ell, u_\ell, v_\ell\}_{\ell \in [L]}}(x) = h_L$, where the successive post-activations $\{h_\ell\}_{\ell \in [L]}$ are defined by $h_0 = x$

$$h_{\ell+1} = b^{\ell+1} h_\ell + \frac{u^{\ell+1}}{\sqrt{d}} \sigma_{\ell+1} \left( \frac{(v^{\ell+1})^\top h_\ell}{\sqrt{d}} \right). \tag{81}$$

In simple words, this multi-layer model simply results from the composition of multiple DAE layers (3). We denote $r_\ell$ the width of layer $\ell$, so that $u_\ell, v_\ell \in \mathbb{R}^{d \times r_\ell}$. We again consider the limit $d \to \infty$, while $r_\ell = \Theta_d(1)$ for all $\ell \in [L]$. We anticipate that the analysis of the online SGD training dynamics trained with online SGD

$$b_{\mu+1}^\ell - b_\mu^\ell = - \frac{\eta}{d^2} \left( \partial_{b^\ell} \mathbb{E}_t \left\| x_1^\mu - f_{\{b_\mu^\ell, u_\mu^\ell, v_\mu^\ell\}_{\ell \in [L]}} \left( \alpha_t x_0^\mu + \beta_t x_1^\mu \right) \right\|^2 \right), \tag{82}$$

$$u_{\mu+1}^\ell - u_\mu^\ell = - \eta \nabla_{u^\ell} \mathbb{E}_t \left\| x_1^\mu - f_{\{b_\mu^\ell, u_\mu^\ell, v_\mu^\ell\}_{\ell \in [L]}} \left( \alpha_t x_0^\mu + \beta_t x_1^\mu \right) \right\|^2 - \eta \frac{\lambda}{d} u_\mu, \tag{83}$$

$$v_{\mu+1}^\ell - v_\mu^\ell = - \eta \nabla_{v^\ell} \mathbb{E}_t \left\| x_1^\mu - f_{\{b_\mu^\ell, u_\mu^\ell, v_\mu^\ell\}_{\ell \in [L]}} \left( \alpha_t x_0^\mu + \beta_t x_1^\mu \right) \right\|^2 - \eta \frac{\lambda}{d} v_\mu. \tag{84}$$

should once again be amenable to theoretical analysis, and shown to converge to a set of deterministic limiting ODEs, borrowing the same technical ideas as previously. We highlight some of the differences in the computations. Massaging the update equations (82) reveals that they depend as previously in (18) on random variables of the form $\lambda_{u,\ell}^1 = (u^\ell)^\top x^1/\sqrt{d}, \lambda_{v,\ell}^1 = (v^\ell)^\top x^1/\sqrt{d}, \lambda_{u,\ell}^0 = (u^\ell)^\top x^0/\sqrt{d}, \lambda_{v,\ell}^0 = (v^\ell)^\top x^0/\sqrt{d}$, over which the integrals appearing in the final expressions bear. The updates will also involve further summary statistics of the type $Q_\tau^{vv,\ell,\ell'}(c) = (v_\tau^\ell)^\top \Sigma(c) v_\tau^{\ell'}/d, Q_\tau^{uv,\ell,\ell'}(c) = (u_\tau^\ell)^\top \Sigma(c) v_\tau^{\ell'}/d, Q_\tau^{uu,\ell,\ell'}(c) = (u_\tau^\ell)^\top \Sigma(c) u_\tau^{\ell'}/d$, and $\mathcal{Q}_\tau^{vv,\ell,\ell'} = (v_\tau^\ell)^\top v_\tau^{\ell'}/d, \mathcal{Q}_\tau^{uv,\ell,\ell'} = (u_\tau^\ell)^\top v_\tau^{\ell'}/d, \mathcal{Q}_\tau^{uu,\ell,\ell'} = (u_\tau^\ell)^\top u_\tau^{\ell'}/d$, which will appear in the end

result, and whose dynamics are tracked by a set of limiting ODEs. Importantly, we thus trade matrix $r \times r$ summary statistics for tensors of size $L \times L \times r \times r$, thus resulting in more cumbersome equations. We leave for future work a thorough theoretical analysis of this model, and the study of the effect of deeper architectures on the generated density.

**Attention models –** Transformer architectures [74] are also often used to parametrize diffusion models. Consider $d = Ld'$ dimensional data $x$, divided in $L$ patches $x_1, ..., x_L \in \mathbb{R}^{d'}$ and reshaped into a sequence $\text{seq}(x) \in \mathbb{R}^{L \times d'}$. Consider the folowing attention-based denoiser:

$$f_w(x) = \text{softmax}\left(\text{seq}(x)ww^\top \text{seq}(x)^\top\right)\text{seq}(x), \tag{85}$$

parametrized by $w \in \mathbb{R}^{d \times r}$ trained on the denoising loss $\mathbb{E}_t\|f_w(\alpha_t x_0 + \beta_t x_1) - \text{seq}(x_1)\|^2$. Let us further assume that all patches $x_\ell$ for $\ell \in [L]$ are drawn from a generalized Gaussian mixture of the form (8), with the different latent variables $c_\ell$ allowed to be correlated across rows — thus reprising, and slightly extending, the data model in [19]. In the asymptotic limit $d' \to \infty, L, r = \Theta_{d'}(1)$, this simple model, close to that considered in [20], and is expected to be once more amenable to analysis using the same set of technical tools, in an approach analogous in spirit to that of [6]. Let us mention that more complex (e.g. deep) architectures are also anticipated to be analyzable, leveraging similar ideas to e.g. [73].

We remark that, on a fundamental level, the anticipated analytical tractability of the above models stems from their pertaining to the class of sequence multi-index models introduced in [19]. This formalism allows for an unified analysis of this class of models. A study of online SGD on sequence multi-index models was reported in full generality in the recent work of [51], to which we refer the curious reader. Finally, we note that while we assumed in this work finite-width networks $r = \Theta_d(1)$, the analysis can also in theory be extended to accommodate wider networks. We refer the interested reader to [75] where such an extension is studied for two-layer feedforward networks, and leave this direction for future work.

## A.7 $\tau \to \infty$ limit

The summary statistics describing the weights of the DAE (3) at convergence are captured by the $\tau \to \infty$ solutions of the ODEs of Result 2.1. The value of the skip connection $b_\infty$ at convergence admits the simple closed-form expression

$$b_\infty = \frac{\Lambda \mathbb{E}_t[\beta_t]}{\Lambda \mathbb{E}_t[\beta_t^2] + \mathbb{E}_t[\alpha_t^2]}, \tag{86}$$

as a function of the schedule fonctions $\alpha_t, \beta_t$. Interestingly, the target density only enters this expression through the effective variance $\Lambda$, which provides an intuitive notion of the spread of the data density, without allowing a finer description of its structure – which is captured by the network component of the DAE (3), rather than its skip connection. In contrast, the other summary statistics $Q_\infty, \mathcal{Q}_\infty, M_\infty, G_\infty$ at convergence obey a set of four coupled equations, which generically admit no simple closed-form expression, aside from simple cases. For illustration, we detail in the following subsection such a simple setting (Gaussian mixture target, linear network) where the equations at convergence can be solved in closed-form.

## A.8 Example : linear model, Gaussian mixture

The previous analysis provides a tight characterization of the training dynamics of the non-linear DAE (3) under the SGD dynamics (4). For completeness and intuition, we conclude the present appendix by expounding a special simple case where the limiting ODEs of Result 2.1 admit a compact, closed-form expression, and consider the linear case $\sigma(x) = x, r = 1, p_t = 0$, when learning a target binary Gaussian mixture with isotropic clusters $\rho = \frac{1}{2}\mathcal{N}(\mu, I_d) + \frac{1}{2}\mathcal{N}(-\mu, I_d)$. We assume the squared norm $\|\mu\|^2$ asymptotically concentrates and denote by $\rho$ its limiting value. Finally, we consider the limit of vanishing learning rate $\eta \to 0$. In this limit, the quadractic term $q_{(2)}^{t,c}$, which is of

order $\Theta(\eta^2)$, can be neglected. The limiting ODEs (75) simplify in this case to

$$\frac{d}{d\vartheta}M = -\mathbb{E}_t\left[\beta_t^2 M^2 + \mathcal{Q}(\beta_t^2 + \alpha_t^2) + \mathcal{Q}\left(\alpha_t^2 + \beta_t^2(1+\rho)\right) - 2\left((1-b\beta_t)\beta_t(\rho+1) - \alpha_t^2 b\right) + \lambda\right]M$$

$$\frac{d}{d\vartheta}\mathcal{Q} = -\mathbb{E}_t\left[2\mathcal{Q}\left(\beta_t^2 M^2 + \mathcal{Q}(\beta_t^2 + \alpha_t^2)\right) - 2\left((1-b\beta_t)\beta_t(M^2 + \mathcal{Q}) - \alpha_t^2 b\mathcal{Q}\right) + \lambda\mathcal{Q}\right]. \tag{87}$$

The evolution of the skip connection $b$, on the other hand, remains unchanged from (76). Let us now seek a solution of (87) at convergence, in the limit $\vartheta \to \infty$. It can be straightforwardly verified that,

$$M = \rho\mathcal{Q}, \qquad \mathcal{Q} = \frac{\frac{\mathbb{E}_t[\beta_t]\mathbb{E}_t[\alpha_t^2]}{\mathbb{E}_t[\beta_t^2 + \alpha_t^2]}\rho - \frac{\lambda}{2}}{\mathbb{E}_t[\beta_t^2(1+\rho) + \alpha_t^2]} \tag{88}$$

is a solution of (87) at convergence. Note that the identity $M = \rho\mathcal{Q}$, together with the definitions $M = w^\top\mu/\sqrt{d}$, $\mathcal{Q} = w^\top w/d$, implies that $w$ lies entirely in $\mathrm{span}(\mu)$. In other words, the weights $w$ of the DAE recover perfectly the direction $\mu$ along which the data distribution $\rho$ exhibits non-trivial structure.

Turning to the generative process, we aim at describing the generated density $\hat{\rho}$ in the one-dimensional subspace $\mathcal{E} = \mathrm{span}(\mu)$, in which the original distribution $\rho$ exhibits non-trivial structure. The SDE (12) describing the evolution of the projection of a sample in this subspace is for the considered setting linear, and can be solved in closed form as

$$Z_t = Z_0 e^{\int_0^t ds(\Delta_s^\infty + \Gamma_s \mathcal{Q})}, \qquad Z_0 \sim \mathcal{N}(0, \mathcal{Q}). \tag{89}$$

In the orthogonal subspace $\mathrm{span}(\mu)^\perp$, the law of a sample is still given by an isotropic Gaussian, as described by equation (13) in Result 2.2. Therefore, the law of a sample $X_t$ remains Gaussian at all times, with the variance in $\mathrm{span}(\mu)$ increasing over sampling time to adapt to – and approximate– the structure of the target distribution $\rho$ in this subspace. This simple linear case sheds light on the workings of the DAE-parametrized diffusion models. Over training, the weights identify and learn the relevant structural features of the target distribution. Subsequently, the learned weights drive the generative process to reproduce the target structure in the identified subspace, while approximating the density in the orthogonal space by an isotropic Gaussian.

## B  Derivation of Result 2.2

In this section, we derive the tight characterization of Result 2.2 for the learnt generative transport process (7).

### B.1  Generative SDE

We remind the generative SDE, leveraged to generate samples from $\hat{\rho}(t)$ starting from $X_0 \sim \mathcal{N}(0, \mathbb{I}_d)$:

$$\frac{dX_t}{dt} = \left(\dot{\beta}_t - \frac{\dot{\alpha}_t}{\alpha_t}\beta_t + \epsilon_t\frac{\beta_t}{\alpha_t^2}\right)f_{b_\tau,w_\tau,v_\tau}(X_t) + \left(\frac{\dot{\alpha}_t}{\alpha_t} - \frac{\epsilon_t}{\alpha_t^2}\right)X_t + \sqrt{2\epsilon_t}dW_t, \tag{90}$$

with $W_t$ a Wiener process and $\epsilon_t$ the diffusion schedule. Introducing the shorthands

$$\Gamma_t = \dot{\beta}_t - \frac{\dot{\alpha}_t}{\alpha_t}\beta_t + \epsilon_t\frac{\beta_t}{\alpha_t^2} \tag{91}$$

$$\Delta_t^\tau = b_\tau\Gamma_t + \frac{\dot{\alpha}_t}{\alpha_t} - \frac{\epsilon_t}{\alpha_t^2}, \tag{92}$$

the generative SDE can be written more compactly as

$$\frac{dX_t}{dt} = \Delta_t^\tau X_t + \Gamma_t\frac{w_\tau}{\sqrt{d}}\sigma\left(\frac{w_\tau^\top X_t}{\sqrt{d}} + p_t v_\tau\right) + \sqrt{2\epsilon_t}dW_t. \tag{93}$$

Importantly, note that the non-linear term $\sigma(\cdot)$ acts on the projection of $X_t$ in the space $\mathcal{W}_\tau$ spanned by the columns of the trained weights matrix $w_\tau$. Furthermore, its image also resides in $\mathcal{W}_\tau$. Thus, the challenging non-linear part of the transport only acts in a small finite-dimensional space, and can be handled carefully in isolation. In contrast, the dynamics in the orthogonal space $\mathcal{W}_\tau^\perp$ is still non-trivially high-dimensional, but simply linear and thus easily to analyzed. This motivates one to examine in succession the variable $Z_t \equiv w_\tau^\top X_t/\sqrt{d}$ and the projection $Y_t \equiv \Pi_{\mathcal{W}_\tau}^\perp X_t$ of $X_t$ in $\mathcal{W}_\tau^\perp$.

## B.2 Dynamics in $\mathcal{W}_\tau$

Let us first ascertain the evolution of $Z_t$, which tracks the evolution of a sample $X_t$ in the weight space $\mathcal{W}_\tau$. It follows directly from (93) that $Z_t$ obeys the $r-$dimensional SDE

$$\frac{d}{dt}Z_t = \Delta_t^\tau Z_t + \Gamma_t \mathcal{Q}_\tau \sigma \left(Z_t + p_t v_\tau\right) + \sqrt{2\epsilon_t}\mathcal{Q}^{1/2}dB_t, \tag{94}$$

with $B_t$ a $r-$dimensional Wiener process, and $\mathcal{Q}_\tau$ the summary statistic sharply characterized in Result 2.1. This recovers equation (12) of Result 2.2.

## B.3 Dynamics in $\mathcal{W}_\tau^\perp$

In $\mathcal{W}_\tau^\perp$, the transport induced by the SDE (93) is simply linear:

$$\frac{dY_t}{dt} = \Delta_t^\tau Y_t + \sqrt{2\epsilon_t}dH_t, \tag{95}$$

with $H_t$ here a $(d-r)-$dimensional Wiener process. This SDE admits a compact closed-form solution

$$Y_t = e^{\int_0^t ds\Delta_s^\tau}Y_0 + e^{\int_0^t ds\Delta_s^\tau}\int_0^t e^{-\int_0^s dh\Delta_h^\tau}\sqrt{2\epsilon_s}dWs. \tag{96}$$

By Itô isometry, $Y_t$ is Gaussian with law

$$Y_t \sim \mathcal{N}\left(0_{\mathcal{W}_\tau^\perp}, e^{2\int_0^t ds\Delta_s^\tau}\left[1 + 2\int_0^t e^{-2\int_0^s dh\Delta_h^\tau}\epsilon_s ds\right]\Pi_{\mathcal{W}_\tau}^\perp\right), \tag{97}$$

which recovers equation (13). This completes the derivation of Result 2.2. $\qquad\square$

## B.4 Discretized sampling

As a final remark, let us note that the derivation presented in the present Appendix can be carried out in completely unchanged fashion starting from any discretization of the generative SDE (1). Let $t_0 = 0, t_1, ..., t_T \in (0, 1)$ and consider the Euler-Mayurama discretization of the stochastic process for $k \in [\![0, T-1]\!]$:

$$\begin{aligned}X_{k+1} - X_k =&(t_{k+1} - t_k)\left[\left(\dot{\beta}_{t_k} - \frac{\dot{\alpha}_{t_k}}{\alpha_{t_k}}\beta_{t_k} + \epsilon_{t_k}\frac{\beta_{t_k}}{\alpha_{t_k}^2}\right)f_{b_\tau, w_\tau, v_\tau}(X_k) + \left(\frac{\dot{\alpha}_{t_k}}{\alpha_{t_k}} - \frac{\epsilon_{t_k}}{\alpha_{t_k}^2}\right)\right]X_k \\ &+ \sqrt{2\epsilon_{t_k}(t_{k+1} - t_k)}\xi_k,\end{aligned} \tag{98}$$

starting from $X_{t_0} \sim \mathcal{N}(0, \mathbb{I}_d)$. In (98), $\xi_k \sim \mathcal{N}(0, \mathbb{I}_d)$ independently for each step $k$. Then the following version of Result 2.2 holds:

**Result B.1. (_Discrete dynamics_)** _Consider a discretization $t_1, ..., t_T \in (0, 1)$ and the discretized sampling process $(X_k)_{k \in [\![1, T]\!]}$ (98). Denote $Y_k = \Pi_{\mathcal{W}_\tau}^\perp X_k$ and $Z_k = w_\tau^\top X_k/\sqrt{d}$, for a process $X_k$ satisfying the generative process (98) from an initialization $X_0 \sim \mathcal{N}(0, \mathbb{I}_d)$. Then $Z_t$ follows the low-dimensional stochastic process_

$$Z_{k+1} - Z_k = (t_{k+1} - t_k)\left[\Delta_{t_k}^\tau Z_t + \Gamma_{t_k}\mathcal{Q}_\tau \sigma\left(Z_k + p_{t_k}v_\tau\right)\right] + \sqrt{2\epsilon_{t_k}(t_{k+1} - t_k)}\mathcal{Q}_\tau^{1/2}\zeta_k, \tag{99}$$

_from an initial condition $Z_0 \sim \mathcal{N}(0, \mathcal{Q}_\tau)$, with $\zeta_k \sim \mathcal{N}(0, \mathbb{I}_r)$ and $\mathbb{E}[\zeta_k\zeta_l^\top] = \delta_{kl}\mathbb{I}_r$. On the other hand, $Y_k$ is independently Gaussian-distributed as_

$$Y_k \sim \mathcal{N}\left(0_{\mathcal{W}_\tau^\perp}, \left[\prod_{j=0}^{k-1}\left(1 + (t_{j+1} - t_j)\Delta_{t_j}^\tau\right)^2 + \sum_{j=0}^{k-2}2\epsilon_{t_j}(t_{j+1} - t_j)\prod_{l=j+1}^{k-1}\left(1 + (t_{j+1} - t_j)\Delta_{t_j}^\tau\right)^2 + 2\epsilon_{t_{k-1}}(t_k - t_{k-1})\right]\Pi_{\mathcal{W}_\tau}^\perp\right). \tag{100}$$

## C   Derivation of Corollary 2.3

Result 2.2 already provides a tight asymptotic characterization of the law of a sample $X_t$ in terms of its projection $Z_t$ (12) in the weights space $\mathcal{W}_\tau$ (characterized by a $r-$dimensional ODE) and its Gaussian component $Y_t$ (13) in the orthogonal space $\mathcal{W}_\tau^\perp$. A weakness of this characterization, however, lies in that it relies on a *training-time dependent* space $\mathcal{W}_\tau$, with respect to which the characterization is formulated. Intuitively, this space rotates and changes as the model is further trained, making the result rather unwieldy. To palliate this shortcoming, one would rather select a *fixed, $\tau-$independent*, reference subspace $\mathcal{E}$ of finite dimension $R = \Theta_d(1)$, and transfer the characterization of Result 2.2 to this fixed subspace. Formally, this means ascertaining the law of the projection of $X_t$ in $\mathcal{E}$, from that of its projections in $\mathcal{W}_\tau, \mathcal{W}_\tau^\perp$. This is made possible because the overlap between the spaces $\mathcal{W}_\tau$ and $\mathcal{E}$ is captured by the summary statistic $G_\tau$, characterized in result 2.1. Formalizing this agenda is the object of the present Appendix.

Let us fix an orthonormal basis $\{e_j\}_{j=1}^R$ of $\mathcal{E}$, stacked vertically in the matrix $E \in \mathbb{R}^{d \times R}$. We remind that we aim at characterizing the law of $E^\top X_t$. To that end, for any $1 \leq j \leq R$, start from the decomposition

$$e_j^\top X_t = (\Pi_{\mathcal{W}_\tau} e_j)^\top (\Pi_{\mathcal{W}_\tau} X_t) + e_j^\top \Pi_{\mathcal{W}_\tau}^\perp Y_t, \tag{101}$$

where we decomposed $X_t$ into its projections in $\mathcal{W}_\tau, \mathcal{W}_\tau^\perp$. Note that, from Result 2.2 the two terms of this decomposition are independent. In the following, we sequentially ascertain the distribution of each of the terms in the decomposition (101).

### C.1   Law of $(\Pi_{\mathcal{W}_\tau} e_j)^\top (\Pi_{\mathcal{W}_\tau} X_t)$

To compute $(\Pi_{\mathcal{W}_\tau} e_j)^\top (\Pi_{\mathcal{W}_\tau} X_t)$, we first aim to decompose $e_j, X_t$ in a basis of $\mathcal{W}_\tau$. Let us consider the eigendecomposition of the summary statistic $\mathcal{Q}_\tau = w_\tau^\top w_\tau / d$ (characterized in Result 2.3) as

$$\mathcal{Q}_\tau = U_\tau S_\tau U_\tau^\top. \tag{102}$$

This means that $B_\tau = 1/\sqrt{d}(S_\tau^+)^{1/2} U_\tau^\top w_\tau^\top$ forms a set of $r$ orthonormal vectors (or a set of orthonormal vectors plus zero vectors if $\mathcal{Q}_\tau$ is rank deficient), which we will use as a basis. We denoted $S_\tau^+$ the Moore-Penrose pseudo-inverse of $S_\tau$. The components of the reference vectors $E \in \mathbb{R}^{d \times R}$ (with columns $\{e_j\}$) and $X_t$ in this basis are then given by

$$B_\tau E = \frac{1}{\sqrt{d}} (S_\tau^+)^{1/2} U_\tau^\top w_\tau^\top E = (S_\tau^+)^{1/2} U_\tau^\top G_\tau^\top \tag{103}$$

$$B_\tau X_t = \frac{1}{\sqrt{d}} (S_\tau^+)^{1/2} U_\tau^\top w_\tau^\top X_t = (S_\tau^+)^{1/2} U_\tau^\top Z_t, \tag{104}$$

where $Z_t$ is characterized in Result 2.2. Then, very simply, the decomposition of $X_t$ in the reference basis $E$ restricted to $\mathcal{W}_\tau$ reads

$$(\Pi_{\mathcal{W}_\tau} e_j)^\top (\Pi_{\mathcal{W}_\tau} X_t) = e_j^\top B_\tau^\top B_\tau X_t = G_\tau \mathcal{Q}_\tau^+ Z_t \tag{105}$$

### C.2   Law of $E^\top \Pi_{\mathcal{W}_\tau}^\perp Y_t$

In distribution, $E^\top \Pi_{\mathcal{W}_\tau}^\perp Y_t$ inherits the Gaussianity of $Y_t$, as established in Result 2.2. It has mean zero and covariance

$$e^{2 \int_0^t ds \Delta_s^\tau} E^\top \Pi_{\mathcal{W}_\tau}^\perp E = e^{2 \int_0^t ds \Delta_s^\tau} E^\top (\mathbb{I}_d - B_\tau^\top B_\tau) E$$

$$= e^{2 \int_0^t ds \Delta_s^\tau} \left[ \mathbb{I}_R - G_\tau \mathcal{Q}_\tau^+ G_\tau^\top \right]. \tag{106}$$

## C.3 Law of $E^\top X_t$

One is now in a position to ascertain the law of $E^\top X_t$. Putting the above results together, in distribution:

$$E^\top X_t \stackrel{d}{=} G_\tau \mathcal{Q}_\tau^+ Z_t + \mathcal{N}\left(0_R, e^{2\int_0^t ds \Delta_s^\tau}\left[\mathbb{I}_r - G_\tau \mathcal{Q}_\tau^+ G_\tau^\top\right]\right), \tag{107}$$

which recovers Corollary 2.3. $\qquad\qquad\square$

## D  Additional experiments

### D.1  Additional details on the numerical experiments

In this Appendix, we provide further specifications on the numerical experiments illustrated in Fig. 7, 3 and Fig. 4.

**Generative process–**  In all the figures, the sampling was carried out by discretizing the interval $(0, 1)$ in $N$ steps $t_k = 1/N$ for $k \in [\![0, N]\!]$, and running the discretized SDE (98) in experiments, and the associated theoretical characterization of Results B.1 and 2.3 for the theoretical predictions, up to a stopping time $0 \leq t_f \leq 1$. In Fig. 7, $N = 100, t_f = 0.95$; in Fig. 8, $N = 100, t_f = 0.98$ and in Figs. 3,?? and 4, $N = 50, t_f = 0.98$. Note that all choices for $N, t_f$ are up to the experimentalist, and captured by the theoretical characterizations. Generically, one needs to opt for $t_f < 1$ due to the DAE-parametrized SDE (7) being ill-defined at $t = 1$, since $\alpha_1 = 0$. This is an artifact of the neural network parametrization; the ground-truth SDE (1) is on the hand well-defined even at $t = 1$.

**Discretization of the manifold density $\pi$–**  In the generic case where $\pi(\cdot)$ (8) is not discrete, the ODE updates (10) still involve an integral over $d\pi(c)$, with $c$ spanning $\mathbb{R}^\kappa$. For instance, in the setting of Fig. 4, at generation $g = 2$, $\kappa = r = 2$ and $\pi(c) = \Pi_{\mathcal{W}_\tau^{(1)}} \hat{\rho}^{(1)}(c)$. The latter is however still characterized in terms of a SDE (12), and not in closed-form. As a first step, we thus generated $4000$ samples from $\pi$, using the theoretical characterization of Result 2.2, and approximated the density using the `scipy` [78] implementation of Gaussian kernel density estimation (KDE), in order to access a smooth estimation of $\pi$. The bandwidth was elected to be $1.5$ times that determined using the Silverman method [67]. To perform the integral with measure $d\pi(c)$, we discretized $\pi$ over a $10 \times 10$ grid, restricting the support to $[-1.5, 1.5] \times [-2.5, 2.5]$ where almost all of its mass was found to lie. The relative weights of the $10 \times 10 = 100$ discretized points were then evaluated from the KDE estimation, and overall normalization was finally enforced to ensure the relative weights sum to $1$. Finally, this discretization was used in evaluating the theoretical characterization of Result 2.1, replacing the integrals over $\pi$ by finite sums over the $100$ points of the discretization. All results have been observed to be rather robust with respect to the choice of discretization, range, and bandwidth.

**Preprocessing of the MNIST images–**  Finally, we detail the procedure used to evaluate the covariance of MNIST sevens used in Fig. 3. The total MNIST training set was used, retaining only sevens. The data was vectorized (flattened), centered, and normalized by $300$. The empirical covariance was finally evaluated over the entire dataset, and used to generate the Gaussian target density considered in Fig. 3.

**Evaluation of the Hellinger distance–**  To estimate the Hellinger distance between $\hat{\rho}$ and $\rho$ (see Fig. 2 (right)), we first sample $5000$ points from the trained diffusion model, project them in the considered subspace $\mathcal{E}$, and approximate the density using the `scipy` implementation of Gaussian KDE. In the case of Fig. 2 (right), we used $\mathcal{E} = \text{span}(\mu)$, and used a $1000$-points grid discretization of the interval $[-10, 10]$ for the purpose of the KDE. The Hellinger distance between $\rho$ and the (KDE of) $\hat{\rho}$ is then numerically estimated as

$$H(\Pi_\mathcal{E}\rho, \Pi_\mathcal{E}\hat{\rho}) = \int_\mathcal{E} dz \left[\sqrt{(\Pi_\mathcal{E}\rho)(z)} - \sqrt{(\Pi_\mathcal{E}\hat{\rho})(z)}\right]^2. \tag{108}$$

By the same token, the theoretical prediction of the Hellinger distance is obtained by sampling $5000$ samples from the theoretical expression for $\hat{\rho}$, as described in Result 2.3, and repeating the same KDE procedure.

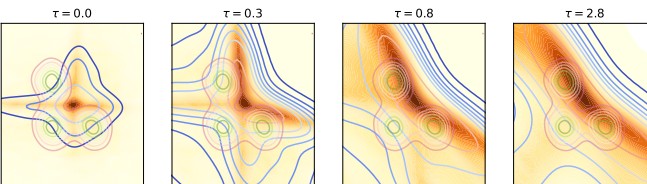

Figure 7: Evolution of the projected density $\Pi_{\mathcal{E}}\hat{\rho}_\tau$ generated by a DAE (3) with $r = 4$ hidden units and $\sigma = \mathrm{ReLU}$ activation, trained on a trimodal Gaussian mixture, with $\eta = 0.2, \lambda = 1.5, \epsilon_t = 0.1, , p_t = 0, \alpha_t = 1 - t, \beta_t = t, \mathcal{G} = \{0.7\}$, from a warm start. The generative SDE (7) was run up to $t = 0.98$, and the subspace $\mathcal{E}$ is spanned by the centroids of the target density. Different panels correspond to different training times $\tau$. Blue contours: contour levels of the theoretical prediction of Corollary 2.3 for the density $\Pi_{\mathcal{E}}\hat{\rho}_\tau$. Colormap: numerical experiments in large but finite dimension $d = 1000$. Green contours: contour levels of the target density $\rho$. Over training time, the four branches of the generated density rotate to align with the clusters of the target density, with two branches merging in the process.

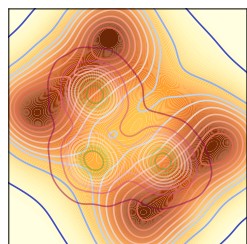

Figure 8: Density $\Pi_{\mathcal{E}}\hat{\rho}_\tau$ generated by a DAE (3) with $r = 2$ hidden units and $\sigma = \tanh$ activation, trained on a trimodal Gaussian mixture, with $\eta = 0.2, \lambda = 1.5, \epsilon_t = 0.1, p_t = 0, \alpha_t = 1 - t, \beta_t = t, \mathcal{G} = \{1/2\}, \tau = 2.8$. The generative SDE (7) was run up to $t = 0.98$, and the subspace $\mathcal{E}$ is spanned by the centroids of the target density. Blue contours: contour levels of the theoretical prediction of Corollary 2.3 for the density $\Pi_{\mathcal{E}}\hat{\rho}_\tau$. Colormap: numerical experiments in large but finite dimension $d = 1000$. Green contours: contour levels of the target density $\rho$.

### D.2 Additional examples

**Trimodal Gaussian mixture —** For completeness, we conclude this Appendix by illustrating the theoretical results 2.1 2.3 on an additional example, namely a trimodal Gaussian mixture density with isotropic clusters.

We consider a generative model parametrized by a DAE (3) with $r = 4$ hidden units and ReLU activation. Fig. 7 illustrates, for different training times $\tau$, the generated density $\hat{\rho}_\tau$ projected in the space $\mathcal{E}$ spanned by the cluster centroids of the target density. A comparison between the theoretical predictions (blue contour levels) and numerical experiments in large but finite dimension $d = 1000$ (orange colormap) reveals a good agreement. Interestingly, the modes of the generated density $\hat{\rho}_\tau$ rotate over training time to align with the modes of the target density $\rho$, with two modes merging in the process. The resulting density $\hat{\rho}_\tau$ at large training time $\tau$ exhibits a similar geometry to the target density $\rho$, without however perfectly reproducing it – a sign of the architectural bias due to the limited expressivity of the model (3), which cannot perfectly generate the target distribution.

Perhaps unsurprisingly, this bias furthermore strongly depends on the architecture of the DAE. Fig. 8 represents the density generated by a DAE with $r = 2$ hidden units and tanh activation, for the same target density $\rho$, with all parameters otherwise unchanged, revealing a very different geometry compared to the ReLU network. In particular, the model fails to generate a trimodal density, with four modes emerging instead. This instance of architectural bias can be easily rationalized. Observe indeed that from equation (12) of Result (2.2), for odd activations such as $\sigma = \tanh$, the transport process is equivariant with respect to the transformation $X \rightarrow -X$. In other words, the generated

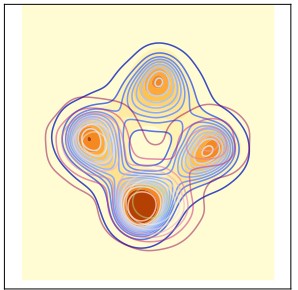

Figure 9: Evolution of the projected density $\Pi_{\mathcal{E}}\hat{\rho}_\tau$ generated by a DAE (3) with $r = 2$ hidden units and $\sigma = \tanh$ activation, trained on a trimodal Gaussian mixture, with $\eta = 0.5, \lambda = 0.1, \epsilon_t = 0.0,, p_t = \cos(\pi t), \alpha_t = 1 - t, \beta_t = t, \mathcal{G} = \{0.2, 0.4, 0.6, 0.8\}$, from a random initialization. For two unit vectors $e_1, e_2$, the target density is the mixture $\rho = {}^1/2\mathcal{N}(-3e_2, I_d) + {}^1/6\mathcal{N}(3e_1, I_d) + {}^1/3\mathcal{N}(-3e_1, I_d)$. The generative SDE (7) was run up to $t = 0.9$, and the subspace $\mathcal{E}$ is spanned by $e_1, e_2$. Different panels correspond to different training times $\tau$. Blue contours: contour levels of the theoretical prediction of Corollary 2.3 for the density $\Pi_{\mathcal{E}}\hat{\rho}_\tau$. Colormap: numerical experiments in large but finite dimension $d = 1000$.

density $\hat{\rho}_\tau$ then necessarily exhibits a symmetry with respect to inversions around the origin —thus forbidding the existence of an odd number of modes. This provides a particularly simple yet telling example of how the choice of architecture can strongly constrain the geometry of the generated densities.

**Class imbalance —** The above example concerns a balanced Gaussian mixture, with all three clusters sharing equal relative probability $^1/3$. One may naturally wonder whether the model can also adapt to class imbalance. Fig. 9 is set for the same target density as Fig. 8, with the difference that clusters now have probabilities $^1/2, ^1/3, ^1/6$. While the generated density still presents a spurious mode (see discussion above), it correctly reproduces the clusters, and correctly gives higher mass to the most probable clusters.

**Fashion MNIST —** We finally provide another example on a real dataset, namely FashionMNIST [87]. This dataset corresponds to $28 \times 28$ gray-scale pictures of clothing items; for simplicity, we only retain images of t-shirts (class 0) and dresses (class 3). We plot in Fig. 10 the density produced by a model parametrized by a $r = 2$ DAE, alongside the target density. The blue curves on the other hand represent theoretical predictions for a target density corresponding to a single Gaussian whose covariance is given by the empirical covariance of the original dataset. Importantly, in contrast to the MNIST experiment illustrated in Fig. 3 in the main text, the numerical experiment were directly conducted on the original dataset, rather than a Gaussian approximation thereof. Furthermore, two classes were kept, instead of a single one in Fig. 3. Fig. 10 shows that the theoretical prediction still reasonably captures the shape of the generated density, and notably once more exhibits a visible reduction in variance compared to the target density.

### D.3 The effect of time encoding

We conclude this appendix by discussing the effect of including the time encoding $p_t$ through its associated set of weights $v$ in the DAE model (3). For the binary target mixture described in the main text (see Fig. 2), we plot in Fig. 2.1 the evolution of the summary statistics $M, \mathcal{Q}$ of Result 2.1 over training time, alongside that of the skip connection strength $b$ and encoding weights $v$. The two plots correspond to a model with no time encoding (i.e. $p_t = 0$) and a model endowed with a sinusoidal time encoding $p_t = \cos(\pi t)$. As can be observed, the introduction of the time encoding has a very small effect, and the curves are left sensibly unchanged. The generated densities, illustrated in Fig. 12, are also strongly similar. A more quantitative viewpoint is displayed in Fig. 13, which shows how the introduction of a time encoding yields a slightly lower, but overall very similar, Hellinger distance between target and generated densities at large training times. These observations temptingly suggest

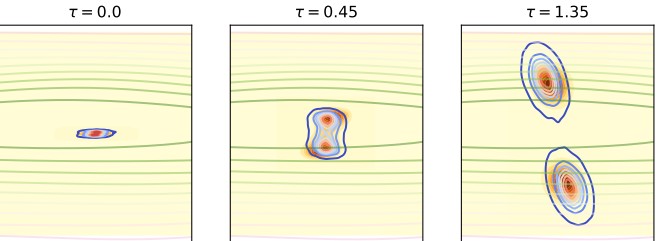

Figure 10: (Evolution of the density $\Pi_{\mathcal{E}}\hat{\rho}_\tau$ generated by a DAE (3) with $r = 2$ hidden units and $\sigma = \tanh$ activation, trained on a the original FashionMNIST [87] dataset, with $\eta = 0.2, \lambda = .784, \epsilon_t = p_t = 0, \alpha_t = 1 - t, \beta_t = t, \mathcal{G} = \{1/2\}$, namely the same parameters as Fig. 3. The generative SDE (7) was run up to $t = 0.98$, and the subspace $\mathcal{E}$ is spanned by principal components of the target density. Different panels correspond to different training times $\tau$. Blue contours: contour levels of the theoretical prediction of Corollary 2.3 for the density $\hat{\rho}_\tau$. Colormap: numerical experiments on the FashionMNIST dataset. Green contours: contour levels of the target density $\rho$.

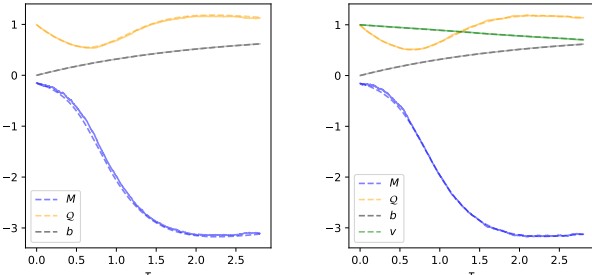

Figure 11: Evolution of the summary statistics $M_\tau, \mathcal{Q}_\tau$ and of the skip connection strength $b_\tau$ and time encoding weights $v_\tau$ as a function of the training time $\tau$, for $\sigma = \tanh, r = 1, \alpha_t = 1 - t, \beta_t = t, \mathcal{G} = \{0.2, 0.4, 0.6, 0.8\}$. The target density the same bimodal Gaussian mixture as Fig. 2. Solid lines: numerical experiments in dimension $d = 1000$. Dashed: theoretical characterization (10) of Result 2.1. **Right**: no time encoding $p_t = 0$. **Left**: with a time encoding $p_t = \cos(\pi t)$. The introduction of a time encoding leaves the training dynamics sensibly unchanged.

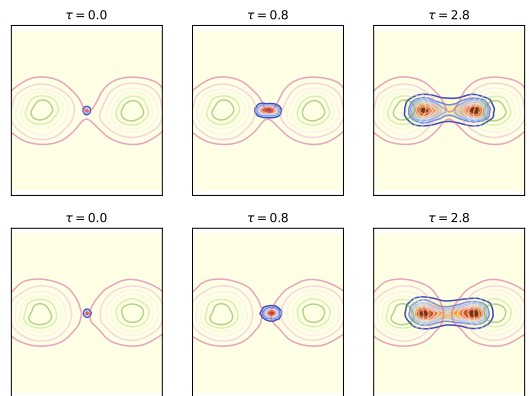

Figure 12: Evolution of the projected density $\Pi_{\mathcal{E}}\hat{\rho}_\tau$ generated by a DAE (3) with $r = 1$ hidden unit and $\sigma = \tanh$ activation, trained on a bimodal Gaussian mixture, with $\eta = 0.2, \lambda = 1.5, \epsilon_t = 0, \alpha_t = 1 - t, \beta_t = t, \mathcal{G} = \{0.2, 0.4, 0.6, 0.8\}$. The generative SDE (7) was run up to $t = 0.9$, and the subspace $\mathcal{E}$ is a plane containing the centroid of the target density. Different panels correspond to different training times $\tau$. Blue contours: contour levels of the theoretical prediction of Corollary 2.3 for the density $\Pi_{\mathcal{E}}\hat{\rho}_\tau$. Colormap: numerical experiments in large but finite dimension $d = 1000$. Green contours: contour levels of the target density $\rho$. **Top**: no time encoding $p_t = 0$. **Bottom**: with a time encoding $p_t = \cos(\pi t)$.

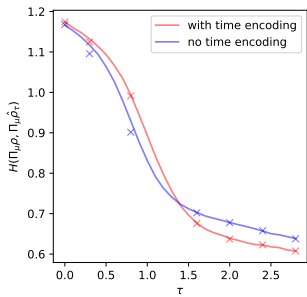

Figure 13: In the same setting as Fig. 12, Hellinger distance between the target and generated densities, projected in the space spanned by the centroid, as a function of the training time $\tau$. **Red**: model with a time encoding $p_t = \cos(\pi t)$. **Blue**: without time encoding $p_t = 0$.

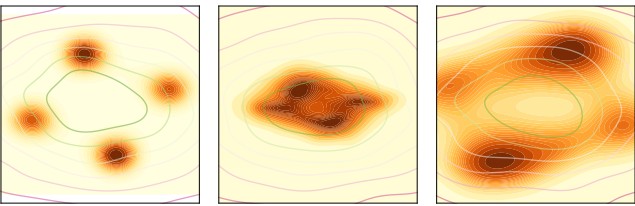

Figure 14: (Evolution of the density $\Pi_{\mathcal{E}}\hat{\rho}_\tau$ generated by a DAE (3) with $r = 2$ hidden units and $\sigma = \tanh$ activation, trained on $5000$ samples of a Gaussian distribution with covariance mathching that of MNIST sevens (see also Fig. 3), with $\eta = 0.2, \lambda = .784, \epsilon_t = p_t = 0, \mathcal{G} = \{0, 1/7, 2/7, 3/7, 4/7, 5/7, 6/7\}$. The generative SDE (7) was run up to $t = 0.98$, and the subspace $\mathcal{E}$ is spanned by principal components of the target density. (**left**) $\alpha_t = 1 - t, \beta_t = t$ (**middle**) $\alpha_t = \cos(\pi t^4/2), \beta_t = \sin(\pi t^4/2)$ (**right**) $\alpha_t = 1 - t, \beta_t = t$ and the skip connection strength is untrained and fixed at $b = 0.75$. Colormap: numerical experiments. Green contours: contour levels of the target density $\rho$.

that, in all probed settings, for this simple model, the inclusion of a time encoding has an overall small effect on the qualitative behavior of the considered model.

### D.4 Influence of the schedule and skip connection on the generated density

In the main text, we described how the limited expressivity of the considered DAE architecture could be conducive to a phenomenon akin to mode collapse, where the generated density displays a largely reduced variance, when compared to the target density. This failure mode could in turn lead to model collapse, namely the rapid degradation of the successive generated densities when samples produced by the model are re-used for training. In this subsection of Appendix D, we briefly mention some parameters of the problem that qualitatively influence the shape of the generated density, and may influence the mode collapse phenomenon. Fig. 14 reprises the experiment on the Gaussian approximation of MNIST sevens of Fig. 3 (reminded on the leftmost panel). In the middle panel, with all other parameters otherwise unchanged, the interpolation schedule is changed from the linear schedule $\alpha_t = 1 - t, \beta_t = t$ to a more intricate cosine schedule $\alpha_t = \cos(\pi t^4/2), \beta_t = \sin(\pi t^4/2)$. This change of schedule is accompanied by a visible shift in the structure of the generated density when all other parameters are kept fixed. Notably, the latter displays a less pronounced multimodal structure. In the rightmost panel, for the linear schedule, we illustrate an experiment where the skip connection is kept fixed at $b = 0.75$ and left untrained. Visibly, the generated density sees its variance increase. Qualitatively, a higher value of $b$ translates into a density with larger spread. These preliminary experiments hint at the role of many of the parameters of the problem in shaping the generated density. A precise and quantitative description of such effects warrants further theoretical and empirical studies, and we leave such questions for future work.

