# OpenReview forum: "A solvable model of learning generative diffusion: theory and insights"
_NeurIPS.cc/2025/Conference — NeurIPS 2025 poster_

### Official Review · Reviewer_WWBX · 2025-06-29

**Clarity:** 3
**Significance:** 3
**Originality:** 4
**Rating:** 4
**Confidence:** 2

**Summary:**

This paper presents a theoretical analysis on the training dynamics for diffusion-based generative models. The authors examine a simple network architecture--two-layer denoising auto-encoder--trained with online stochastic gradient descent on high-dimensional data exhibiting a low-dimensional manifold structure. A key contribution is the derivation of a tight asymptotic characterization of the low-dimensional projections of samples generated by the learned model, detailing its dependence on the number of training samples. Building on this, the work discusses the emergence of mode collapse and its potential to lead to model collapse upon retraining with synthetic data.

**Questions:**

1. I am not convinced that the ODE of $\mathcal{Q}_\vartheta$, defined in Eq.10, is relied on $G_\vartheta$. According to the definition of $\mathcal{Q}_\tau$ in Eq.9, it is totally determined by $w_\tau$ and $d$. However, the definition of $G_\tau$ depends on the matrix $E$, which is arbitrarily chosen and independent with $w_\tau$. Could the authors further clarify the dependence of $\mathcal{Q}_\vartheta$ on the matrix $E$? Similar questions also exist for $Q_\vartheta$ and $M_\vartheta$.

**Ethical Concerns:**

["NO or VERY MINOR ethics concerns only"]

**Final Justification:**

The paper has made some progress of theoretical importance, lacking of architectural generality. After the disscussion during the rebuttal, I will not chage my rate.

**Limitations:**

yes

**Paper Formatting Concerns:**

Yes

**Quality:**

4

**Strengths And Weaknesses:**

- Strengths
1. Interesting Theoretical Results: The paper delivers insightful asymptotic results on the projection of the stochastic process. This can advance the understanding of learning dynamics in diffusion models.
2. Strong Experimental Validation: The theoretical analysis is well-supported by experimental verification, demonstrating a strong coincidence between the theoretical predictions and empirical observations.
3. Valuable Insights into Collapse Phenomena: The paper provides a very interesting and valuable explanation of the mechanisms underlying mode collapse and its progression to model collapse, which are crucial failure modes in generative models.

- Weaknesses
1. Limited Architectural Generality: While the theoretical insights are significant, their direct applicability is currently limited to a two-layer auto-encoder architecture. It would substantially enhance the paper's impact if these findings could be extended or shown to generalize to more complex network architectures commonly used in practice.

---

> ### Author Rebuttal · Authors · 2025-07-29
>
> We thank the reviewer for their reading of our work and their pertinent comments.
>
> > Limited Architectural Generality: While the theoretical insights are significant, their direct applicability is currently limited to a two-layer auto-encoder architecture. It would substantially enhance the paper's impact if these findings could be extended or shown to generalize to more complex network architectures commonly used in practice.
>
> We acknowledge that these simplifications constitute an important limitation of current theoretical analyses of diffusion models. Reaching a theory capturing the full complexity of practical diffusion models is obviously an extremely ambitious and difficult goal, that will likely only be gradually reached through many successive papers by the research community, advancing the theoretical understanding step by step. We hence feel it makes sense to evaluate our paper not by how far its stands from modern practice, but rather by how it advances this line of theoretical research. In this light, our paper was the first to report a full characterization of the generated density, considerably refining many previous results.
>
> As we discuss, the architecture considered already prescinds a number of unrealistic assumptions (such as infinite width or sparse weights) considered in a number of prior works. We agree with the reviewer that generalizing such results to more complex architectures is a crucial research direction. As a first step, we anticipate that the analysis can be extended to multilayer architectures, and architectures including self-attention layers, provided all the weights remain low-rank in the considered asymptotic limit, using the same technical approach. While a careful exploration of such architectures warrants separate analyses and are left for future work, we will outline how they can be analyzed in the revision of this manuscript.
>
> Finally, large-width networks should also be amenable to being studied building on the ideas of [a], as we discuss in our answer to Reviewer pjCW. Result 2.1 would need to be adapted similarly to [a]. Results 2.2 and 2.3 directly carry through to this case.
>
> [a] Veiga et al, Phase diagram of Stochastic Gradient Descent in high-dimensional two-layer neural networks, 2022
>
>
>
> >I am not convinced that the ODE of $Q_\theta$ should depend on $G_\theta$
>
> The reviewer is right that the ODE for $Q_\theta$ does not depend on $G_\theta$, for the reason they outline. We chose to write $ \frac{d}{d\theta} Q_\theta =F_Q(O_\theta)$, not making explicit which arguments $F_Q$ actually depends on, merely for notational compactness and convenience. We will clarify this point in the revised manuscript.

---

> > ### Comment · Reviewer_WWBX · 2025-08-04
> >
> > Thanks the authors for the explanation. I have no further questions.

---

### Official Review · Reviewer_pjCW · 2025-06-30

**Clarity:** 3
**Significance:** 3
**Originality:** 3
**Rating:** 4
**Confidence:** 3

**Summary:**

This paper presents a solvable theoretical model for studying diffusion-based generative learning. It analyzes a two-layer denoising autoencoder trained via SGD on high-dimensional Gaussian mixtures with a low-dimensional manifold structure. The authors derive sharp asymptotic characterizations of the training dynamics and the generated density. The analysis captures both successful learning and failure modes, such as mode collapse and model collapse, offering insights into the inductive bias of DAE-based generative models.

**Questions:**

1. While the model studied is analytically tractable, it differs from practical diffusion architectures (See the weakness). Can the authors comment more explicitly on how the insights generalize to more realistic settings?

**Ethical Concerns:**

["NO or VERY MINOR ethics concerns only"]

**Final Justification:**

1. The paper provides a technically rigorous and well-executed analysis.
2. While the assumptions are standard in theory papers, they may limit the direct applicability of the results to practical, large-scale models.

**Limitations:**

Yes.

**Paper Formatting Concerns:**

NA.

**Quality:**

3

**Strengths And Weaknesses:**

Strengths:
1. The paper provides a technically rigorous and well-executed analysis.
2. The paper is generally well written and structured.
3. The work advances theoretical understanding of how generative diffusion models learn from data. The treatment of architectural bias and re-training effects across generations appears novel and impactful.

Weakness:
1. While the assumptions are standard in theory papers, they may limit the direct applicability of the results to practical, large-scale models. First, in the training of foundation models, the number of samples is much larger than the data dimension. Second, the width of hidden layers is much larger than the input layer.
2. Certain derivations, particularly in the appendix, are quite dense and may be challenging to follow without strong background knowledge. The paper would benefit from additional high-level explanations or intuitive summaries. In addition, the discussion of empirical results could be expanded slightly to aid readers less familiar with theory.

---

> ### Author Rebuttal · Authors · 2025-07-29
>
> We thank the reviewer for their reading of our work and insightful questions, which we answer below.
>
> >While the assumptions are standard in theory papers, they may limit the direct applicability of the results to practical large-scale models. First, in the training of foundation models, the number of samples is much larger than the data dimension. Second, the width of hidden layers is much larger than the input layer.
>
> We acknowledge that these simplifications constitute an important limitation of current theoretical analyses of diffusion models. Reaching a theory capturing the full complexity of practical diffusion models is obviously an extremely ambitious and difficult goal, that will likely only be gradually reached through many successive papers by the research community, advancing the theoretical understanding step by step. We hence feel it makes sense to evaluate our paper not by how far its stands from modern practice, but rather by how it advances this line of theoretical research. In this light, our paper was the first to report a full characterization of the generated density, considerably refining many previous results.
>
> As we discuss, the architecture considered already prescinds a number of unrealistic assumptions (such as infinite width or sparse weights) considered in a number of prior works. We strongly agree with the reviewer that pushing those analyses to cover more realistic models constitutes an important future research direction.
>
> Addressing more in detail the reviewer's concerns, we note that the large data regime can be accessed by simply taking the limit $\tau\to\infty$ in our theoretical expressions. We will add a more detailed discussion of this limit in the revised version. The results can also be extended to large-width networks. Result 2.1 would need to be adapted similarly to [a]. Results 2.2 and 2.3 directly carry through to this case. We chose to focus on the small-width cases $r=O(1)$ so as to reach simpler characterizations, fully in terms of finite-dimensional quantities. We will discuss this extension in the revised version of the manuscript.
>
>
> [a] Veiga et al, Phase diagram of Stochastic Gradient Descent in high-dimensional two-layer neural networks, 2022
>
>
> >Certain derivations, particularly in the appendix, are quite dense and may be challenging to follow without strong background knowledge. The paper would benefit from additional high-level explanations or intuitive summaries. In addition, the discussion of empirical results could be expanded slightly to aid readers less familiar with theory.
>
> We will include more discussions to aid comprehension and build intuition, in between the technical segments, to enhance the readability of the Appendix.
>
> >While the model studied is analytically tractable, it differs from practical diffusion architectures (See the weakness). Can the authors comment more explicitly on how the insights generalize to more realistic settings?
>
> We acknowledge the considered architecture is very simplified. On the other hand, it includes a number of the features of practical (e.g. U-Net) models, such as bottleneck layers and a skip connection. The insights on the respective role of these components, e.g. the discussion on the role of the skip connection on model collapse in section 4, will hopefully pave the way towards building insight on the role of these components in more complex models. Such a study would warrant detailed empirical work which fall out of the scope of the present manuscript -- whose main goal was to derive  a tight characterization of the generated density, advancing a line of prior theory works.
>
> We finally anticipate that more complex architectures, including multilayer and attention models, can be analyzed using the same technical approach, provided all weights remain low-rank in the considered high-dimensional limit. While a careful exploration of these more realistic models would warrant separate studies, we will outline how they could be analyzed leveraging similar technical ideas in the revised manuscript.

---

> > ### Comment · Reviewer_pjCW · 2025-08-04
> >
> > Thanks for your response. Could you present some discussions to aid comprehension and build intuition here?

---

> > > ### Author Response · Authors · 2025-08-04
> > > **Further intuition on the derivation of the technical results**
> > >
> > > The driving idea behind the derivation is the following.
> > >
> > > - The goal is to characterize the generated density, and the generative flow, in the subspace of interest $E$. Since the flow is driven by the DAE weights $w$, one needs a characterization of the components $G$ of $w$ in the subspace $E$.
> > >
> > > - To characterize these components, one must turn to study the training dynamics. Writing down the SGD iteration (18), one can see the evolution of the components $G$ over training depend on the components $M$ of $w$ in the space spanned by the target distribution, and on the second order statistics $Q$ of the weights. The object of Appendix A -- and Theorem 1-- thus consists in ascertaining the joint evolution of $M,G,Q$ under the SGD evolution. This is carried out in Appendix A, until subsection A.4, where the SGD iterations are rewritten in terms of $M,Q,G$, and closed on these variables. A notable point is that in the limit of high dimensions, these quantities concentrate in probability, and are thus described by a set of deterministic ODEs (10).
> > >
> > > - Now that $G,Q$ are determined, one can turn to study the generative flow. First, we project the flow onto the subspace spanned by the weights (subsection B.2), and in its orthogonal space (subsection B.3). These projected flows are completely governed by $Q$, which was already characterized. Finally, the dynamics are further projected in the subspace  of interest $E$, in eqs. (96-99). These projected dynamics make the components $G$ appear.
> > >
> > > Hopefully, this brief sketch of the intuition underlying the derivation helps clarify and motivate the technical sections. An expanded version of this paragraph will be included at the relevant locations in the Appendix, to clarify to non-expert readers the purpose of the main derivation steps, and guide them through the Appendix.

---

> > > > ### Comment · Reviewer_pjCW · 2025-08-05
> > > >
> > > > Thanks for the explanation. I will maintain my positive score.

---

### Official Review · Reviewer_Lq8Y · 2025-06-30

**Clarity:** 2
**Significance:** 3
**Originality:** 3
**Rating:** 4
**Confidence:** 3

**Summary:**

This paper presents a fully tractable theoretical analysis of a simplified diffusion‐based generative model. Concretely, the authors study a two-layer denoising autoencoder (DAE) trained by online SGD on high-dimensional target distributions given by Gaussian mixtures supported on a low-dimensional manifold. The main contributions are:
1. Training dynamics (Result 2.1): In the limit $d,n\to\infty$ with $n/d=\Theta(1)$, the SGD evolution of a finite set of summary statistics of the weights is shown to follow a closed system of deterministic ODEs (Eqns (10), (11)).

2. Generative sampling (Result 2.2): Using the trained DAE parameters, the sampling SDE is reduced to a low-dimensional nonlinear SDE in the learned subspace plus an analytic Gaussian in the orthogonal complement (Eqns (12), (13)).

3. Projected density (Corollary 2.3): Combining these, the law of any fixed low-dimensional projection of the generated samples admits a fully explicit characterization (Eqn (14)).

4. Insights into failure modes: The framework predicts and quantitatively explains mode collapse (loss of diversity) and model collapse (amplified collapse under retraining on synthetic data) for DAEs on realistic targets (Sections 4, Fig. 4).

5. Empirical validation: Numerical simulations on synthetic Gaussian mixtures and MNIST-covariance targets confirm the tight match between theory and finite-ddd experiments (Figs. 2–3, 7–8).

**Questions:**

1. Can the methodology be adapted to multi-layer or even UNet architecture? What are some of the technical obstacles?

2. How sensitive are the results if the target density exhibits heavy tails for non-diagonalizable covariances?

**Ethical Concerns:**

["NO or VERY MINOR ethics concerns only"]

**Final Justification:**

While the current paper could benefit from a cleaner presentation, with fewer technical details and more insights and intuition behind the derived results in the main text, I do believe the technical contribution is solid hence recommend the paper for acceptance.

**Limitations:**

Unsure what practical insights we can gain from theoretical investigations like this. The authors should more clearly motivate why such an analysis is useful. Right now a characterization is given and then mode(l) collapse is given as an example demonstration of the usefulness of such characterization, but it feels lacking. More example applications/analyses should be given to justify the usefulness of such theoretical characterization. Moreover, what theoretical insights became available based on this particular characterization that were previously not available in prior studies of more unrealistic assumptions?

**Quality:**

3

**Strengths And Weaknesses:**

**Strengths**
The paper provides a rigorous comprehensive theoretical characterization of 2-layer DAE with skip connection, including both learning and generation dynamics for a finite-dimension network that is more realistic than prior works, assuming infinite dimension or trivial dimension. The theoretical results shed light on the diminishing variance over training leading to mode collapse and also model collapse iteratively training on generated data. The methodology employed here can be in principle applied to analyze more complicated architectures as well.

**Weaknesses**
1. While the theoretical results agree well with the experiments, both the architecture and the data distribution here still remain simplified for realistic cases.

* As the authors pointed out themselves, the architecture analyzed here is oversimplified (2-layer DAE). While the analytical model is able to predict diminishing variance leading to mode(l) collapse, it does not provide us further insights into some of the more puzzling phenomena in large text-conditioned image-generative diffusion models, such as compositionality.

* Moreover, the data distribution analyzed here is quite simple. The analysis relies on joint diagonalizability of cluster covariance and may not extend well to non-Gaussian, heavy-tailed or multimodal distributions beyond mixtures.

2. The demonstrated agreement with experiment is shown for synthetic and toy examples. Even with the MNIST data, it is only tested on one of the digit classes, 7, which is also distributed based on Gaussian distribution.

3. It is unclear what practical insights the theoretical characterization provides, besides the predicted mode(l) collapse, especially considering the toy and simplified setting of the derived analytical model. Here a few points might be interesting to theoretically explore further:

* The authors have explained a bit about the functionality of the skip connection in preventing mode(l) collapse. A further characterization of the role of skip connection might be useful.

* The authors have briefly explained that the time embedding has minimal impact on the theoretical results presented. In text-conditioned image generative models, conditional text embeddings are included in the same way as the positional embedding of time. It could be interesting to analyze/characterize the role of conditioning cues.

4. How does theoretical insights gained from the analytical model inform us of practical architecture improvements or training strategies? For example, can we adjust the skip-connection schedules to mitigate mode collapse?

5. In general in terms of presentation, while I appreciate that the authors try to provide intuition behind each equation and derivation, the notations become quickly out-of-hand and hard to keep track of beyond equation 14. I would suggest rather than presenting these as the main results of the paper, move them to the appendix and shifting the focus to the insights that can be gained from these derived results and how they are informative practically.

---

> ### Author Rebuttal · Authors · 2025-07-29
>
> We thank the reviewer for their detailed reading of our manuscript and many constructive comments.
>
> >While the theoretical results agree well with the experiments, both the architecture and the data distribution here still remain simplified for realistic cases.
>
> We acknowledge that these simplifications constitute an important limitation of current theoretical analyses of diffusion models. Reaching a theory capturing the full complexity of practical diffusion models is obviously an extremely ambitious and difficult goal, that will likely only be gradually reached through many successive papers by the research community, advancing the theoretical understanding step by step. We hence feel it makes sense to evaluate our paper not by how far its stands from modern practice, but rather by how it advances this line of theoretical research. In this light, our paper was the first to report a full characterization of the generated density, considerably refining many previous results.
>
>  As we discuss in l.126-136, the architecture considered already prescinds a number of unrealistic assumptions (such as infinite width or sparse weights) considered in some prior works.
> We further anticipate that the analysis can be extended to more complex architectures, including multilayer and attention models, provided all weights remain low-rank in the considered high-dimensional limit. While a careful exploration of these more realistic models would warrant separate studies, we will outline how they could be analyzed leveraging similar technical ideas in the revised manuscript.
>
> Finally, the data distribution considered is sizably more general than linear subspaces [12,14,48]  or finite Gaussian mixtures [18] often considered previously.
>
> >As the authors pointed out themselves, the architecture analyzed here is oversimplified (2-layer DAE). While the analytical model is able to predict diminishing variance leading to mode(l) collapse, it does not provide us further insights into some of the more puzzling phenomena in large text-conditioned image-generative diffusion models, such as compositionality. [...] Moreover, the data distribution analyzed here is quite simple. The analysis relies on joint diagonalizability of cluster covariance and may not extend well to non-Gaussian, heavy-tailed or multimodal distributions beyond mixtures.
>
> The considered data distribution already constitutes an improvement compared to previous theoretical works that consider less realistic linear subspaces [12,14,48]  or finite Gaussian mixtures [18] distributions. This paper is furthermore the first theoretical work reporting analytical predictions displaying a tight quantitative agreement with simulations for realistic data distributions, such as the one illustrated in Fig. 3 (right). Prescinding data distribution assumptions necessary in theory works is a gradual process, and we also hope that the gap between theory works and real data distributions will be further bridged by future works by us or other researchers.
>
> We note that the joint diagonalizability of clusters is a common assumption in theoretical analyses, considered for instance in [18,24], or [a]. The considered data distribution in the present work, which encompass a manifold structure and generic (jointly-diagonalizable) covariance is more general. As we already mention in l.168, it also covers as a special case a large class of heavy-tailed distributions, as some heavy-tailed distributions can be expressed as an infinite superposition of Gaussians.
>
> Beyond the jointly-diagonalizable case, we also expect to be able to analyze the setting where each cluster covariance has  eigenvectors in generic positions, mathematically modeled by independent Haar matrices. We will dicuss this possible extension in the revised manuscript.
>
> [a] Wu et al, Theoretical Insights for Diffusion Guidance: A Case Study for Gaussian Mixture Models, 2024
>
> >The demonstrated agreement with experiment is shown for synthetic and toy examples. Even with the MNIST data, it is only tested on one of the digit classes, 7, which is also distributed based on Gaussian distribution.
>
> To the best of our awareness, this paper is the first theoretical work reporting analytical predictions displaying a tight quantitative agreement with simulations for realistic data distributions. Note that the Gaussian approximation of MNIST 7s still yields realistic images very similar to the original distribution, see Fig. 3 (right). We will include further experiments in the revised manuscript, including on the original MNIST 7 images, mixtures of several MNIST digits, and other simple image datasets.
>
>
> >It is unclear what practical insights the theoretical characterization provides, besides the predicted mode(l) collapse, especially considering the toy and simplified setting of the derived analytical model. [...] How does theoretical insights gained from the analytical model inform us of practical architecture improvements or training strategies? For example, can we adjust the skip-connection schedules to mitigate mode collapse?
>
>
> We acknowledge that similarly to many papers in the theory of diffusion models, e.g. those discussed around l.66, the aim of this paper is primarily mathematical in scope -- namely, deriving a tight analytical characterization of the generated density. On the other hand, the paper furthermore provides a number of insights on the role of the architectural components, notably on the role of the skip connection in shaping the final density, discussed in section 4. Following the reviewer's question, we furthermore conducted preliminary experiments that suggest that freezing the skip connection (rather than training it), and varying the interpolation schedule (e.g. $\alpha_t=\cos(2t/\pi),\beta_t=\sin(2t/\pi)$), may help mitigating the collapse in the examples discussed in the main text. We will include further discussion of these cases, along their theoretical analysis, in the revised version of the manuscript.
>
>
>
> >The authors have explained a bit about the functionality of the skip connection in preventing mode(l) collapse. A further characterization of the role of skip connection might be useful.
>
>
> Following the reviewers comment, we conducted supplementary experiments, freezing the skip connection to a fixed value and leaving it untrained, in order to more finely ascertain its role. These preliminary simulations suggest that depending on the value it is fixed at, freezing the skip connection can mitigate (for large skip connection values) or aggravate (for smaller values) the model collapse. Notably, fixing $b=0$ corresponds to a DAE architecture without skip connection, which we will add as an insightful baseline to  shed further light on the role of the skip connection.
>
> >The authors have briefly explained that the time embedding has minimal impact on the theoretical results presented. In text-conditioned image generative models, conditional text embeddings are included in the same way as the positional embedding of time. It could be interesting to analyze/characterize the role of conditioning cues.
>
> We fully agree with the reviewer that the study of conditional models constitutes an exciting direction. However, their study would warrant the development of new mathematical ideas and models that fall out of the scope of the present work.
>
> >In general in terms of presentation, while I appreciate that the authors try to provide intuition behind each equation and derivation, the notations become quickly out-of-hand and hard to keep track of beyond equation 14. I would suggest rather than presenting these as the main results of the paper, move them to the appendix and shifting the focus to the insights that can be gained from these derived results and how they are informative practically.
>
> We thank the reviewer for this suggestion. In the revised version, we will move some of the most notation-heavy technical results to the appendix for better readability of the main text.
>
> >Unsure what practical insights we can gain from theoretical investigations like this. The authors should more clearly motivate why such an analysis is useful. Right now a characterization is given and then mode(l) collapse is given as an example demonstration of the usefulness of such characterization, but it feels lacking. More example applications/analyses should be given to justify the usefulness of such theoretical characterization.[...] Moreover, what theoretical insights became available based on this particular characterization that were previously not available in prior studies of more unrealistic assumptions?
>
> While prior studies report bounds on some probability metrics or distances (e.g. KL divergences) between the target and generated distributions, our work refines these results to a full characterization of the generated density, allowing in particular the elucidation of its structure. This allows in particular the study of mode collapse --- which cannot be fully described by the characterization of a simple metric.

---

> > ### Comment · Reviewer_Lq8Y · 2025-08-02
> >
> > I appreciate the authors' thoughtful responses. However, I still feel like a bit is lacking in the main takeaway from your analyses. Could you maybe summarize what insights (ideally practical) that we can gain from your models/analyses that were previously not known?

---

> > > ### Author Response · Authors · 2025-08-04
> > > **Insights from our analysis**
> > >
> > > As a work inscribed in the field of the theory of diffusion models, the main contribution is technical in nature and aimed at advancing the mathematical understanding of diffusion models. Specifically, the technical contributions include the derivation of a tight characterization of the generated density, which could not be accessed in previous work.
> > >
> > > This characterization allows the elucidation of the structure of the generated density, and comes with some new qualitative insights. Most notably
> > >
> > > - (Role of the skip connection) The skip connection scales as the mean data covariance eigenvalue after training (equation (11), discussed in section 4). For real data distributions, this value is typically small. Due to the role of the skip connection in the generative diffusion, this can conduce to mode collapse. As indicated in the rebuttal, freezing the skip connection may palliate this issue.
> > >
> > > - (Role of the activation) We discuss in appendix D the role of the activation : for a same multimodal target density, a DAE with ReLU activation succeeds in reproducing the multimodal structure (Fig. 7), while an odd activation such as tanh imposes symmetries in the generative flow that result in the generation of spurious modes (Fig. 8). This simple example sheds light on the sizable impact of the choice of the activation function on the generated density.
> > >
> > > - (Emergence of structure over training time) The generated density only acquires the correct structure over training time. If the training is insufficient or stopped too early, the generated density might display erroneous structure, e.g. unimodal instead of bimodal (Fig. 2), or 4 modes instead of 3 (Fig. 7).
> > >
> > > As a final remark, we hope the present paper would be judged in the same light as the line of theory works it advances, namely for its technical contribution in furthering the mathematical understanding of a multifaceted problem coupling learning and generative dynamics. Similarly to many previous works, its primary aim is not to provide immediately applicable guidelines to practitioners -- although we strongly agree with the reviewer that such applications would be very valuable.

---

> > > > ### Comment · Reviewer_Lq8Y · 2025-08-04
> > > >
> > > > I have raised my rating but I do hope that the authors consider improving the presentation of the main results, namely presenting more intuition and insights of behind the derived results in the main text.

---

### Official Review · Reviewer_ecs1 · 2025-07-02

**Clarity:** 4
**Significance:** 3
**Originality:** 3
**Rating:** 4
**Confidence:** 5

**Summary:**

This paper analyzes a solvable flow or diffusion-based generative model, focusing on a two-layer auto-encoder trained with online stochastic gradient descent on high-dimensional data with a low-dimensional manifold structure. The authors provide a precise asymptotic characterization of low-dimensional projections of the generated sample distribution, highlighting its dependence on the number of training samples. The analysis also explains how mode collapse can occur and lead to model collapse when retraining on synthetic data.

**Questions:**

Does the analysis reported in this work provide any insights to general applications of diffusion and flow-based models?
For example, what kind of stratygies might help mitigating the problem of model collapse?

**Ethical Concerns:**

["NO or VERY MINOR ethics concerns only"]

**Limitations:**

Yes.

**Paper Formatting Concerns:**

No major concerns.

**Quality:**

3

**Strengths And Weaknesses:**

The major strength of this work is a rigorous analysis of a solvable flow or diffusion-based generative model. The authors provide a precise asymptotic characterization of low-dimensional projections of the generated sample distribution. The results are interesting and help better understand the properties of diffuion models from a theoretical perspective. Since it is difficult to analyze the diffuion models, the results from
this work is interesting.

The main weakness is that it is not clear what insights the analysis can provided to practical implementations of diffucion and flow-based models.

---

> ### Author Rebuttal · Authors · 2025-07-29
>
> We thank the reviewer for their reading of our manuscript and pertinent questions, which we answer below.
>
> >The main weakness is that it is not clear what insights the analysis can provided to practical implementations of diffucion and flow-based models. Does the analysis reported in this work provide any insights to general applications of diffusion and flow-based models? For example, what kind of stratygies might help mitigating the problem of model collapse?
>
>  Following a large majority of works in the theory of diffusion models, including for instance the related papers cited l.66, our paper primarily aims at understanding from a mathematical viewpoint how a diffusion model can learn the structure of the data distribution over training, and the effect of the model architecture and number of samples. While this field of research does not offer immediately transferable applications, we believe it constitutes a crucial endeavor to ensure a more principled usage and future development of such models. Our work advances this research endeavor, by being the first to our awareness to provide a full tight characterization of the generated density.
>
>  The paper furthermore provides a number of insights that may offer some practical intuition, notably the role of the skip connection in shaping the final density, which we discuss in section 4. Following the reviewers' comments, we conducted further experiments that suggest that freezing the skip connection (rather than training it), and adopting alternative interpolation schedules (e.g. $\alpha_t=\cos(2t/\pi),\beta_t=\sin(2t/\pi)$), may help mitigating the collapse in the discussed tasks. We will include further discussion of these cases, along their theoretical analysis, in the revised version of the manuscript.

---

> > ### Comment · Reviewer_ecs1 · 2025-08-09
> >
> > I appreciate authors' response to my comments on the paper. I confirm that I will keep the oriignal score.

---

### Decision · Program_Chairs · 2025-09-17

**Decision:**

Accept (poster)

**Comment:**

This paper presents mostly theoretical contributions to the understanding of diffusion of flow-based generative models. It uses a highly simplified architecture, which, while not resembling the actual types of architectures used in practice, does remove some of the more unrealistic assumptions in other works (e.g. infinite width assumptions), and incorporates some realistic elements (e.g. skip connections). The results provide an asymptotic characterization of low-dimensional projections of the generated sample distribution.

The reviewers all entered a final score of “weak accept”, indicating that the paper has solid theoretical contributions, but is limited in scope and practicality. The major issues raised were: the architecture considered is too simplified and not realistic compared to how diffusion models are actually built in practice; the theoretical derivations lack explanation, intuition, and discussion; overall few/no practical insights are given as to how these theoretical results can affect how diffusion models are used in practice. However, the reviewers felt that the theoretical contributions were still valuable despite not having direct practical implications.

Hence, I am recommending to accept this submission, and encourage the authors to include more high level explanation and intuition to complement their dense mathematical push to make the paper more broadly accessible to the NeurIPS audience.